# Genetic circuit characterization by inferring RNA polymerase movement and ribosome usage

Amin Espah Borujeni[1,2], Jing Zhang [1,2], Hamid Doosthosseini [1], Alec A. K. Nielsen[1] & Christopher A. Voigt [1✉]

To perform their computational function, genetic circuits change states through a symphony of genetic parts that turn regulator expression on and off. Debugging is frustrated by an inability to characterize parts in the context of the circuit and identify the origins of failures. Here, we take snapshots of a large genetic circuit in different states: RNA-seq is used to visualize circuit function as a changing pattern of RNA polymerase (RNAP) flux along the DNA. Together with ribosome profiling, all 54 genetic parts (promoters, ribozymes, RBSs, terminators) are parameterized and used to inform a mathematical model that can predict circuit performance, dynamics, and robustness. The circuit behaves as designed; however, it is riddled with genetic errors, including cryptic sense/antisense promoters and translation, attenuation, incorrect start codons, and a failed gate. While not impacting the expected Boolean logic, they reduce the prediction accuracy and could lead to failures when the parts are used in other designs. Finally, the cellular power (RNAP and ribosome usage) required to maintain a circuit state is calculated. This work demonstrates the use of a small number of measurements to fully parameterize a regulatory circuit and quantify its impact on host.

[1] Synthetic Biology Center, Department of Biological Engineering, Massachusetts Institute of Technology, Cambridge, MA 02139, USA. [2] These authors contributed equally: Amin Espah Borujeni, Jing Zhang. ✉email: cavoigt@gmail.com

D ebugging a genetic system is analogous to fixing a watch using only the displayed time, without being able to see the internal mechanisms[1]. Genetic circuits are particularly difficult, as their operation requires coordination between many regulatory parts[2]. Genome-scale technologies are illuminating the inner workings of the cell, including transcription, translation, and chemical composition[3–6]. Computational tools are needed to extract parameters to inform models and to scan for errant features that can be fixed by making mutations or selecting different parts[7,8]. These methods need to scale to larger circuits, corresponding to more parameters, without requiring more experiments.

NOR gates have two input promoters, a repressor that inverts their signal, and an output promoter[9,10]. Sets of repressors have been used to build libraries of NOR gates that can be connected to build circuits[11–16]. The flux of RNA polymerase $J_{RNAP}$ on DNA is a critical measurement for genetic circuit construction[17]. The response function of a gate captures how the output $J_{RNAP}$ changes as a function of the input $J_{RNAP}$. The units of $J_{RNAP}$ have been reported in relative promoter units (RPUs) by comparing the strength of a promoter to that of a reference[17], and single-molecule studies have concluded 1 RPU = 0.019 RNAP/s per promoter[18]. The response functions can be used to predict how two gates will connect, which is the basis for design automation software (e.g., Cello)[14,19].

A circuit consists of a DNA sequence containing a pattern of promoters assigned to each gene. The circuit responds to stimuli through genetic sensors that convert an environmental signal, such as the presence of a small molecule, to a promoter activity ($J_{RNAP}$). The circuit responds to different stimuli by changing state, represented by a different pattern of regulator expression and $J_{RNAP}$ emanating from the circuit's promoters. Therefore, the circuit requires continuous cellular power to store its state[19–23]. Maintaining an RNAP flux requires cellular energy and metabolites[24], and gates require a futile cycle of regulator expression and degradation. Circuit states that place a larger burden on the cell take these resources away from cellular maintenance, thus decreasing growth and incentivizing evolutionary breakage[21,25–34].

Using RNA sequencing (RNA-seq), $J_{RNAP}$ can be simultaneously calculated for all of the promoters in a genetic circuit[8,35]. RNA-seq is performed by isolating RNA, converting it to cDNA, deep sequencing the pool, and then mapping the reads to the genome to create a transcript profile[36]. The level of gene expression can be calculated as the average profile height over its length, reported as FPKM (fragments per kilobase of transcript per million mapped reads). FPKM is an estimate of mRNA transcript copy number and has been converted to absolute units using RNA spike-in standards[7,37,38]. When there is a sharp increase or decrease in RNA-seq transcription profile, the magnitude can be used to infer the strength of a promoter or terminator, respectively[39–47]. When the average cDNA fragment size is large (100–500 nt)[48,49], this complicates the calculation because it obscures nearby promoters and requires a correction factor at the transcript ends[8]. To address this, we use a technique that uses short RNA fragments (<50 nucleotide) and single-end mode sequencing[50].

Cellular translation can be quantified using ribosome profiling, which takes global snapshot of ribosomes on mRNA transcripts[51,52]. Ribosome profiling can be used to calculate translational parts, notably ribosome binding sites (RBSs), and to measure translation start sites, pause sites, coupling in operons, and cryptic translation[7,53–59]. It can be performed simultaneously with RNA-seq, where the cellular RNA content is collected, digested, and the ribosome-protected RNA is fractionated, converted to cDNA, and sequenced[54,55]. A profile is built by mapping ribosome P-sites to the DNA sequence, the height of which is the number of ribosomes at this position (the ribosome occupancy)[54,55]. The average profile height over the gene length (the ribosome density, RD), serves as an estimate for protein expression. It can also be used to calculate the steady-state protein abundance, which can be used to estimate the proteome of the cell[54,55].

In this manuscript, all of the parts in a genetic circuit are characterized by combining data from RNA-seq and ribosome profiling. Design automation (Cello) is used to build a 3-input logic gate consisting of seven NOT/NOR gates[14]. The response of the circuit is as predicted across all combinations of inputs (states). The RNA-seq data are used to visualize the changes in the pattern of $J_{RNAP}$ that occur as the circuit is transitioned between states. Methods are developed to extract the strengths of all genetic parts, including that of promoters, ribozymes, RBSs, and terminators. The response functions of all of the gates in the context of the circuit can be calculated and compared directly to measurements in genetic isolation. Despite functioning as expected, the circuit has many internal errors, including cryptic promoters, transcription attenuation, alternative start sites, failed parts, off-target interactions, and a gate that fails completely. These errors are found to not propagate to circuit failure due to the robustness of this circuit topology to the observed parameter variation and failures in individual gates. In addition, the number of RNAPs and ribosomes used to run the circuit are calculated and it is found to require up to 5% of the cell's transcriptional and translational resources. These techniques offer an unprecedented view into the workings of the circuit, thus empowering future design efforts to increase the scale and precision of artificial regulatory networks in cells.

## Results

**RNAP flux and ribosome usage patterns across the circuit.** We sought to study a circuit that works well; that is, behaves as designed and does not lead to an obvious cellular phenotype. To this end, a Cello-designed E. coli circuit was selected that encodes a 3-input logic operation as one of the top-performing circuits from a set of 60 (Fig. 1a)[14]. It consists of sensors that respond to IPTG/aTc/Ara whose outputs are integrated by seven layered NOT/NOR gates that ultimately control the expression of yellow fluorescent protein (YFP) (Fig. 1b). The 6840 bp circuit is carried on a p15a plasmid and a second pSC101 plasmid contains the output promoter driving yfp (Supplementary Fig. 1). For each permutation of input signal, the YFP response is predicted for a population of cells using a mathematical model that combines the empirical response functions of the gates measured in isolation (Fig. 1c). Grossly, the circuit functions as predicted across all states in terms of when it is on or off, but there are quantitative differences (notably, the +/−/− state). Further, we did not observe large growth defects or genetic instability.

For each circuit state, cells were prepared for RNA-seq and ribosome profiling (Methods). Cultures were grown and the RNA was harvested in exponential phase. The end-enriching method is used to save short RNA fragments for sequencing, resulting in library sizes of 10–45 nucleotides[50]. This was found to be effective at resolving promoters in series, reducing the conflating effect of ribozymes, and lessening transcript end effects (Supplementary Figs. 2 and 3). Problematically, this method preserves tRNAs, which comprise up to 65% of the recovered RNA and can introduce a mapping bias against other genomic regions; thus, these reads were manually removed before calculating the transcript levels (FPKM) of the plasmids or genome (Supplementary Fig. 4) (Methods).

The transcript profile can be used to infer the change in RNAP flux over the circuit DNA. Our approach simplifies promoter activity to the production of a continuous source of RNAPs, essentially time- and population-averaging the experimentally

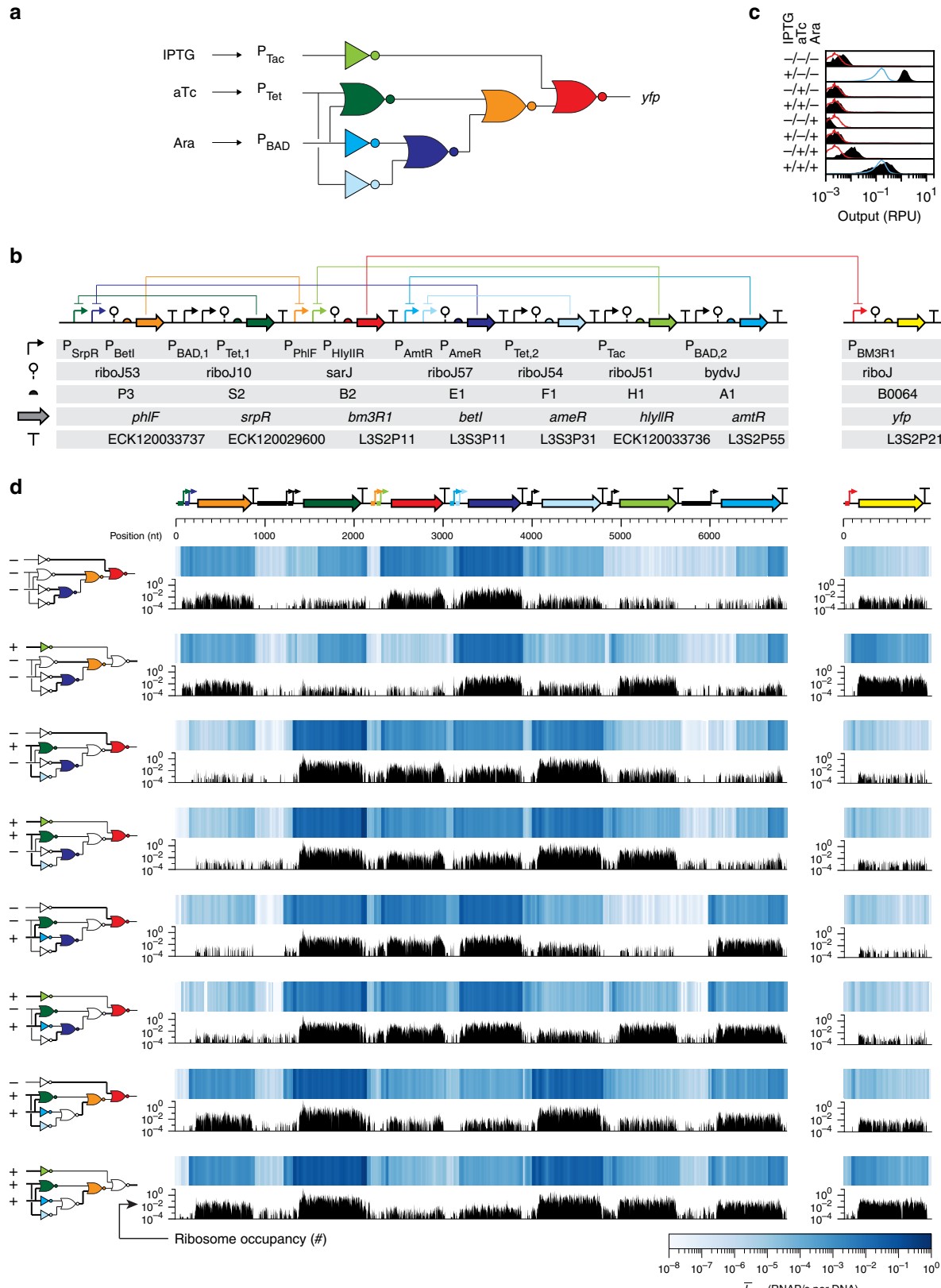

**Fig. 1 Genetic circuit characterization using RNA-seq and ribosome profiling. a** Wiring diagram of the model circuit is shown. The colors correspond to the repressors used for each gate. **b** Genetic parts for the genetic circuit (left) and output promoter (right), carried on different plasmids (Supplementary Fig. 1 and Supplementary Table 5). **c** The measured response of the circuit output to different combinations of inputs (1 mM IPTG, 2 ng/ml (4.32 nM) aTc, and 5 mM Ara). The black distributions are experimental measurements (Methods) compared to Cello predictions (blue if ON, red if OFF). **d** The RNAP flux across the circuit is shown (blue) with the ribosome occupancies (black). Each circuit state is shown, marked by the diagram (thick black lines indicate corresponding wires that are ON; gates are colored if that repressor is being expressed). The conversions of RNA-seq and ribosome profiling data to absolute units are described in the Methods. Source data are provided as a Source Data file.

observed RNAP bursts[60–62]. At steady-state, the flux $J_i$ at each nucleotide $i$ is related to the height of the transcript profile $M_i$ by $J_i = \gamma M_i$, where $\gamma$ is the RNA degradation rate. For the circuit, we assume a constant $\gamma = 0.0067\ \text{s}^{-1}$ for all mRNAs, which is reasonable given that they all encode single repressor genes and are approximately the same length with similar 5′- and 3′-UTR (untranslated region) sequences[8,63]. The profile is often reported in arbitrary units; however, we can convert it to absolute units in two steps (Methods). First, because we know the promoter activities of the circuit output promoter ($P_{BM3RI}$) in RPU (Fig. 1c) we can convert FPKM to RPU (Supplementary Fig. 5). Next, we empirically measured the conversion factor from RPU to RNAP/s for the reference promoter[17] obtained from single-molecule experiments[18]. This flux can be reported in units of $J_{RNAP}$, which is the total flux across all copies of the promoter in the cell, or $\bar{J}_{RNAP}$ which is normalized by the plasmid copy number to give the per promoter flux. The former is required for the response function and the calculations for circuit design, while the latter better captures the flux on a single strand of DNA.

The fluxes on the circuit DNA are shown in Fig. 1d for each state. It is possible to experimentally visualize the signal carrier of a transcriptional circuit as it performs the computation, analogous to watching the flow of electrons in an electronic circuit. The punctate nature of the synthetic circuit can be seen, where repressor genes are strongly turned on and off under the different conditions. This is due to the use of strong promoters and terminators that are spatially colocated. Thus, the appearance of the profile is quite different than that observed for the natural regulatory networks encoded in the genome.

The ribosomes can be visualized in each state of the circuit using ribosome profiling data (Methods). This shows the changes in the ribosome usage for each gate in their on and off states (Fig. 1d). The magnitudes of the bars indicate the number of ribosomes that occur at that position in arbitrary units. Similar to RNA-seq profiles, we can convert these profiles to absolute units. First, these occupancies were normalized by the sum of the occupancies at all positions in the circuit and genome, resulting in a profile that represents the fraction of total active ribosomes in the cell that are recruited to each DNA position. To convert this fraction profile to absolute units, we need an estimate of the total number of active ribosomes in the cell, which has been previously measured for E. coli and is a function of cell growth rate[64]. In our experiments, this corresponds to 20,000 ribosomes that are actively translating mRNAs. The ribosome occupancy pattern across the circuit reflects the one obtained through RNA-seq, as expected (Fig. 1d). However, there is variability in the ribosome usage for each repressor gene, which has implications in the gate function and the cellular resources required to hold its state.

The FPKM and RD measurements for the circuit genes are much higher than for those in the genome (Supplementary Fig. 6). The medians are 220-fold and 140-fold higher, respectively. In addition, we analyzed whether the higher estimated protein expression is dominated by higher transcript levels or higher translation per transcript. Van Oudenaarden and co-workers observed that there is more protein expression noise when the transcript level is low and the translation rate is high[65]. Compared to genomic gene expression, the RD/FPKM levels of circuit proteins are lower, indicating a bias towards lower cell-to-cell variation. Indeed, this was a factor in the selection of these repressors and the design of the gates (Supplementary Fig. 6)[14,16].

**Errors in transcription and translation in the circuit DNA**. The circuit DNA is entirely composed of synthetic parts: promoters are based on well-defined scaffolds, the genes have been codon optimized, the 5′-UTRs contain ribozymes and RBSs that were designed computationally. Still, we observe many examples of transcription and translation differing from that expected (Fig. 2a).

Cryptic promoters can adversely impact circuit function[66,67]. They are identified by calculating the ratio between RNAP flux profiles at neighboring positions $J_{RNAP}(i+1)/J_{RNAP}(i)$ to which a threshold is applied to mark a transcription start site (TSS) (Fig. 2b) (Methods). Using this method, all of the TSSs are identified for the synthetic gate promoters, each of which only has one. However, the location of this TSS is upstream of the annotated position for $P_{SrpR}$, $P_{Tac}$, $P_{BAD,1}$, and $P_{BAD,2}$ (Supplementary Fig. 7). The srpR and amtR genes have strong internal promoters in the sense direction. Remarkably, antisense promoters occur in 6/7 of the repressor genes with some stronger than the forward promoters of the gates. There is also a strong antisense promoter in the middle of both $P_{BAD,1}$ and $P_{BAD,2}$ promoters that corresponds to the known antisense promoter ($P_c$) in $P_{BAD}$[68]. Our method identifies the TSS of this promoter occurring at the 5′-end of the $O_{1L}$ operator. The identified sense and antisense promoters correspond with DNA sequences that are similar to the consensus σ70 binding motif (Supplementary Fig. 8).

All of the gates contain a ribozyme insulator to normalize the 5′-UTR of the transcript. They are all based on variants of the RiboJ insulator, which consists of a hammerhead ribozyme and a 23 nt hairpin added to stabilize ribozyme folding[69]. While the RiboJ sequence varies, this hairpin is always the same sequence, inside of which we discovered a strong antisense promoter at a σ70 binding motif (Fig. 2c and Supplementary Fig. 9). Further, after the upstream promoter, there is a consistent drop in transcription (up to 12-fold) at this antisense promoter. Antisense promoters are known to attenuate RNAP flux from a forward promoter and may lead to better gate performance by lowering basal activity in the off state, which may explain why this hairpin emerged from a screen for parts that improve gate function[67]. Note that this impacts the characterization of the gate response functions and how these data are used by design automation software to connect them (Supplementary Fig. 10).

Transcriptional attenuation is common in long operons, where the early dissociation of RNAP leads to incomplete mRNA transcripts[4,70–75]. We define there being evidence in the profile for attenuation when >10-fold continuous decrease in RNAP flux along the gene sequence is observed. Most of the circuit genes do not show evidence for attenuation, the exceptions being hlyIIR, bm3R1, and yfp (Fig. 2d and Supplementary Fig. 11). In particular, hlyIIR shows a continuous 46-fold decline over the length of the gene. There is an antisense promoter in this gene, but other repressor genes with stronger antisense promoters do not show attenuation. However, the ribosome profiling data show that this promoter also leads to cryptic antisense translation, which is not the case for any of the other cryptic antisense promoters (Fig. 2e). RNAP is coupled to ribosomes and this has been shown to lead to increased collisions with RNAP on the opposite strand, leading to greater attenuation[76–78].

The ribosome profiling data was analyzed to determine the impact of ribosome pausing within genes. We looked for a relationship with codon usage, noting that it has been observed that ribosomes can pause when they encounter rare codons or Shine Dalgarno (SD)-like sequences and this can reduce the expression level[55,79–81]. Rare codons were eliminated when the synthetic repressor genes were codon optimized[16]. Anecdotally, we noticed a rise in occupancy at the 3′-end of bm3R1, which corresponds to strong predicted SD and start codon sequences (Supplementary Fig. 12). However, we could not find a systematic relationship between higher ribosome occupancy and codons that participate in SD-like sequences.

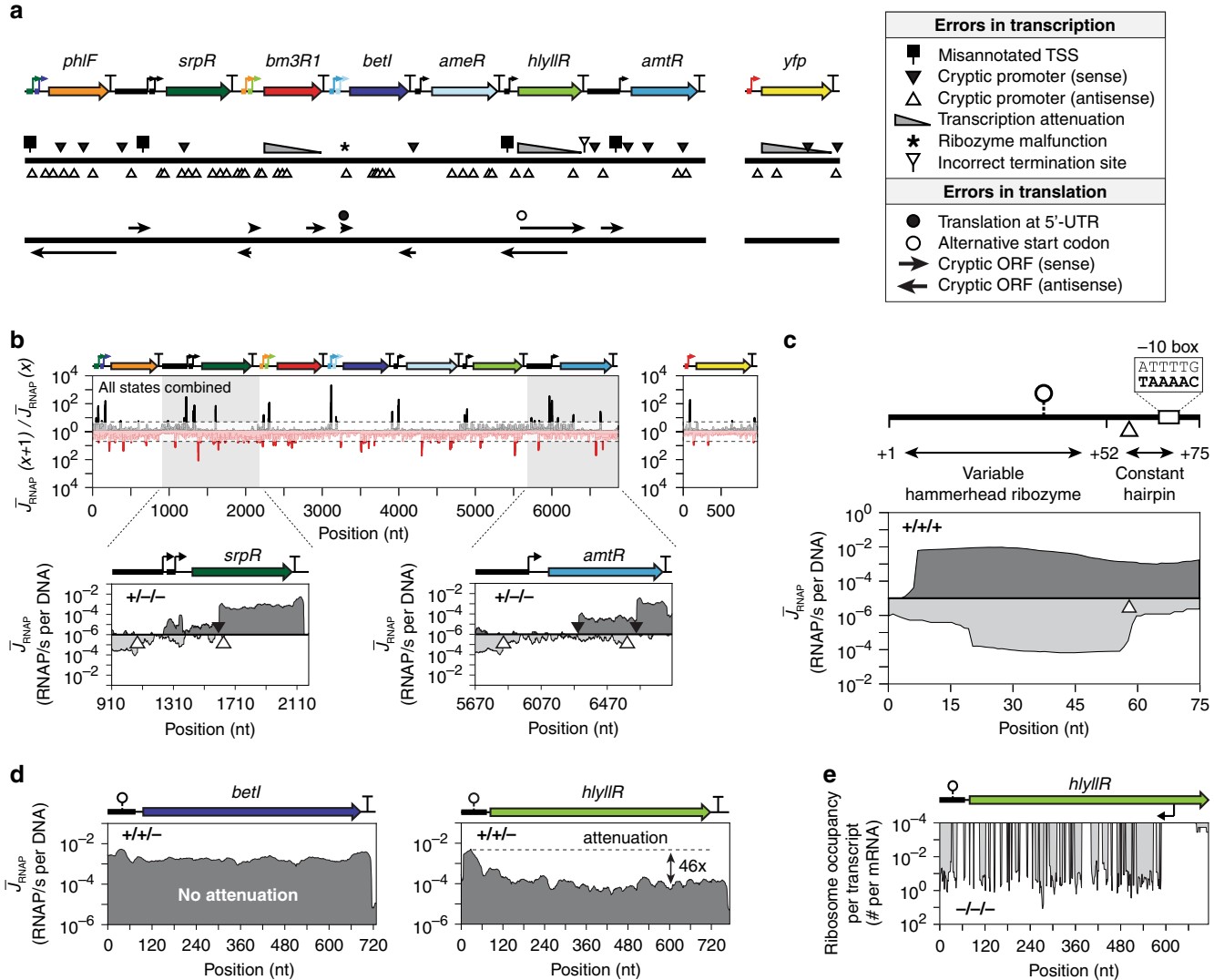

**Fig. 2 Unexpected transcription and translation. a** A summary of transcriptional (top) and translational (bottom) errors, with respect to the designed circuit, are shown. The size of the arrows are accurate to the length of the cryptic ORFs. **b** The detection of TSSs in the sense (top) and antisense (bottom) directions are shown. When the derivative of the profile is larger than the cutoff required to mark a TSS (dashed lines), it is colored darker. The *srpR* and *amtR* transcription units are shown for state +/−/− (IPTG/aTc/Ara). The dark and light gray filled regions represent transcription on sense and antisense strands, respectively. Cryptic promoter positions are marked by filled (sense strand) and empty (antisense strand) triangles. **c** The riboJ53 insulator is shown, marking the unexpected TSS (white triangle) and −10 σ70 motif. The sense and antisense profiles are shown in dark and light gray, respectively. **d** The transcript profiles for the *betI* and *hlyIIR* transcription units is shown (state +/+/−). The dashed line marks the height of the profile at the beginning of the transcript to highlight attenuation. **e** The antisense cryptic promoter is marked and the antisense translation detected through ribosome profiling is shown. Source data are provided as a Source Data file.

**Parameterization of genetic parts**. A complete picture of how parts function in the context of the circuit is illuminated by the RNA-seq and ribosome profiling datasets (Fig. 3, and Tables 1 and 2). Transcriptional parts, including promoters and terminators, can be parameterized using the RNA-seq profile. Parts that operate on the level of mRNA can also be characterized, such as the ribozyme insulators. Ribosome occupancy can be used to calculate the RBS strengths. The transcription and translation of genes can also be determined. Collectively, all 54 genetic parts in the 3-input logic circuit can be fully parameterized with these two experiments. These data can be compared to part measurements made in isolation, usually in a different genetic context, or predicted by computational models.

Previously, we presented methods to extract the promoter strength from the transcript profile by calculating the magnitude of the change before and after a TSS[8,35]. However, the use of large fragment sizes caused biases at either end of the transcript—right at the TSS—and blurred the effects of neighboring promoters or promoters next to ribozymes (Supplementary Fig. 3). Data generated using the end-enriching method allow promoters to be identified automatically and their strength determined with fewer assumptions. The promoter strength is the increase in RNAP flux calculated as the difference between the transcript profile before (averaging positions −20 to −10) and after (averaging positions 10–20) the TSS (Fig. 3a). The promoter strengths were calculated for all 12 promoters, including the sensors and gates, for all the circuit states (Table 1). Cello was then used to predict the strength of all of the promoters in the circuit (Fig. 3b). There is a weak correlation between the measured and predicted strengths, 7% of the promoters fail in at least one state (are off when they should be on and vice versa), and the HlyIIR gate fails completely (discussed below).

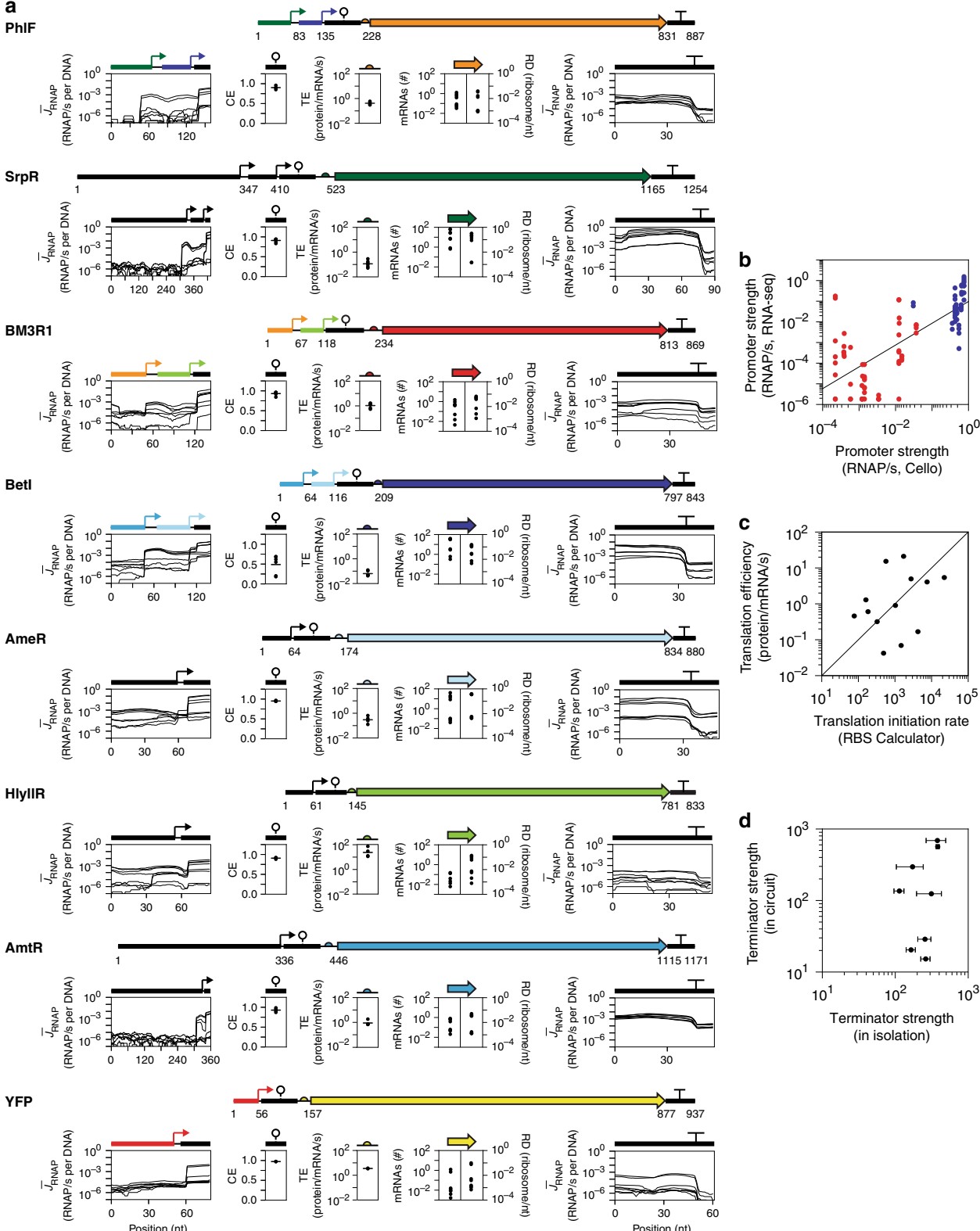

**Fig. 3 Complete characterization of genetic circuit parts. a** The data are organized by the eight transcription units in the circuit. For promoters and terminators, the transcript profiles for all circuit states are shown, with the TSS and TTS marked with an arrow or T, respectively. The ribozyme CE and RBS TE values are shown for the states where the repressor is transcribed (average marked by the horizontal line). The estimated transcript and protein (RD) levels for the genes are shown for the eight induction states. **b** The promoter activities for the promoters as measured by RNA-seq are compared to the values predicted by Cello. Data are shown for all 12 promoters within the circuit for the eight induction states. The points are shown in red if the promoter should be off and blue if it should be on. **c** The TE measured for the circuit genes in the most highly transcribed state are compared against their predicted translation initiation rate (TIR) (Methods). **d** The terminator strengths of all eight terminators as measured using RNA-seq in their most highly transcribed state are compared to their previously measured values in isolation using fluorescence-based assays (data and error bars are from ref. [27], $n \geq 3$ independent measurements on different days). Source data are provided as a Source Data file.

### Table 1 Genetic part parameters.

| Genetic part | Parameter | Units |
|---|---|---|
| **Promoters** | **Strength (×10⁻³)** | **RNAP/s per DNA** |
| | $y_{max,i}$ ($y_{min,i}$)[a] | |
| $P_{AmeR}$ | 67 (0.3188) | |
| $P_{AmtR}$ | 7.4 (0.0002) | |
| $P_{BAD,1}$ | 4.8 (0.0002) | |
| $P_{BAD,2}$ | 6.0 (0.0004) | |
| $P_{BetI}$ | 7.5 (0.0152) | |
| $P_{BM3R1}$ | 9.3 (0.0235) | |
| $P_{HlyIIR}$ | 49 (0.0133) | |
| $P_{PhlF}$ | 0.8 (0.0002) | |
| $P_{SrpR}$ | 0.9 (0.0002) | |
| $P_{Tac}$ | 4.9 (0.0002) | |
| $P_{Tet,1}$ | 171 (0.0030) | |
| $P_{Tet,2}$ | 109 (0.0002) | |
| **Terminators** | **Strength ($T_i$)[b]** | |
| ECK120029600 | 689 | |
| ECK120033736 | 20 | |
| ECK120033737 | 124 | |
| L3S2P11 | 15 | |
| L3S2P21 | 565 | |
| L3S2P55 | 29 | |
| L3S3P11 | 296 | |
| L3S3P31 | 136 | |
| **Ribozymes** | **Cleavage efficiency ($\eta_i$)[c]** | |
| bydvJ | 0.90 | |
| riboJ | 0.89 | |
| riboJ10 | 0.87 | |
| riboJ51 | 0.85 | |
| riboJ53 | 0.91 | |
| riboJ54 | 0.84 | |
| riboJ57 | 0.48 | |
| sarJ | 0.94 | |
| **RBSs** | **Translation efficiency ($\alpha_i$)[d]** | **Protein/s per mRNA** |
| A1 | 0.85 | |
| B2 | 0.57 | |
| BBa_B0064 | 1.22 | |
| E1 | 0.04 | |
| F1 | 0.16 | |
| H1 | 14.5 | |
| P3 | 0.43 | |
| S2 | 0.07 | |

[a]The maximum and minimum promoter strength measured across the eight circuit states. The units are per promoter.
[b]The transcript knockdown that occurs when the upstream promoter is being maximally transcribed across the eight circuit states.
[c]The average cleavage efficiency (CE) calculated across the eight circuit states.
[d]The translation efficiency (TE) measured when the upstream promoter is being maximally transcribed across the eight circuit states. mRNA level is from RNAP flux profile.

### Table 2 Empirical parameterization of gate response functions.

| Promoter | Parameter constant | Units |
|---|---|---|
| | **Binding constant ($k_i$ ×10³)** | **Protein #** |
| $P_{AmeR}$ | 0.55 | |
| $P_{AmtR}$ | 2.24 | |
| $P_{BetI}$ | 0.12 | |
| $P_{BM3R1}$ | 0.43 | |
| $P_{HlyIIR}$ | 9.80 | |
| $P_{PhlF}$ | 0.50 | |
| $P_{SrpR}$ | 0.06 | |
| | **Cooperativity ($n_i$)** | |
| $P_{AmeR}$ | 1.1 | |
| $P_{AmtR}$ | 2.4 | |
| $P_{BetI}$ | 1.7 | |
| $P_{BM3R1}$ | 1.3 | |
| $P_{HlyIIR}$ | 1.7 | |
| $P_{PhlF}$ | 4.2 | |
| $P_{SrpR}$ | 1.2 | |

to be nonfunctional under all conditions. This ribozyme differs from the others in that it is AU-rich and has a smaller folding $\Delta G$ and is thus vulnerable to context effects (Supplementary Fig. 14), and ribosome profiling also reveals it contains an open reading frame (Supplementary Fig. 12).

RBSs recruit ribosomes to the transcript and, indeed, there is an elevated ribosome occupancy in these regions (Figs. 1d and 3a)[52,84]. RBS strengths can be measured empirically by fusing them to a reporter gene[85,86]. However, their strength is highly context-dependent because of sensitivities to 5′-UTR folding and the translation rates of upstream open reading frames[86–89]. Computational models have been developed that capture these effects to predict RBS strength[90,91]. The translational efficiency (TE) of a gene is commonly estimated as the ratio of ribosome density and transcript level[54]. Here, we calculate the TE in absolute units (Methods), which is found to correlate with the translation initiation rate (TIR) predicted by the RBS Calculator (Fig. 3c and Supplementary Fig. 15) (Table 1)[90,91].

Terminators block the progression of RNAP and their strength can be quantified as the magnitude of a drop in the transcript profile (Fig. 3a, Supplementary Fig. 16, and Table 1) (Methods). Previous methods to quantify terminators with RNA-seq data were limited by the large fragment lengths that caused a bias at the 3′-end[8,35]. This is corrected using shorter fragment libraries. These data reveal that the transcription termination site (TTS) does not occur at a precise position; rather, it is distributed across several nucleotides in the poly-T region[27]. The terminator strength $T_S$ is defined as the ratio of the transcript level before the TTS to the level after it[27]. We developed algorithms to identify the TTS region and then calculate $T_S$ using the same size and location for the windows as defined for the promoter calculation (Supplementary Fig. 16) (Methods). Previously, the $T_S$'s were measured for the terminators in isolation using fluorescent reporters[27]. Their quantitative performance differs and some terminators are much weaker in the circuit context (Fig. 3d).

**Gate characterization in the context of a circuit**. The circuit was designed by Cello using gate response functions that were measured independently[14]. This assumes that the gate will produce the same response in the context of the circuit. The response

Surprisingly, despite these failures, the circuit is able to perform the logic operation for which it was designed.

Ribozymes are used to process the transcripts so that those from different TSSs have the same 5′-UTR and increase stability by decreasing RNase affinity[69,82,83]. Self-cleavage releases a small RNA that cannot be seen on the transcript profile when it is smaller than the average fragment size sequenced. To address this, we developed a method by which the ribozyme cleavage efficiency (CE) can be calculated from these data (Supplementary Fig. 13) (Methods). The cleavage efficiencies of most of the ribozymes are high and constant across circuit states, as expected, and are close to the values measured in vitro[14] (Fig. 3a and Table 1). The exception is riboJ57 in the BetI gate, which we find

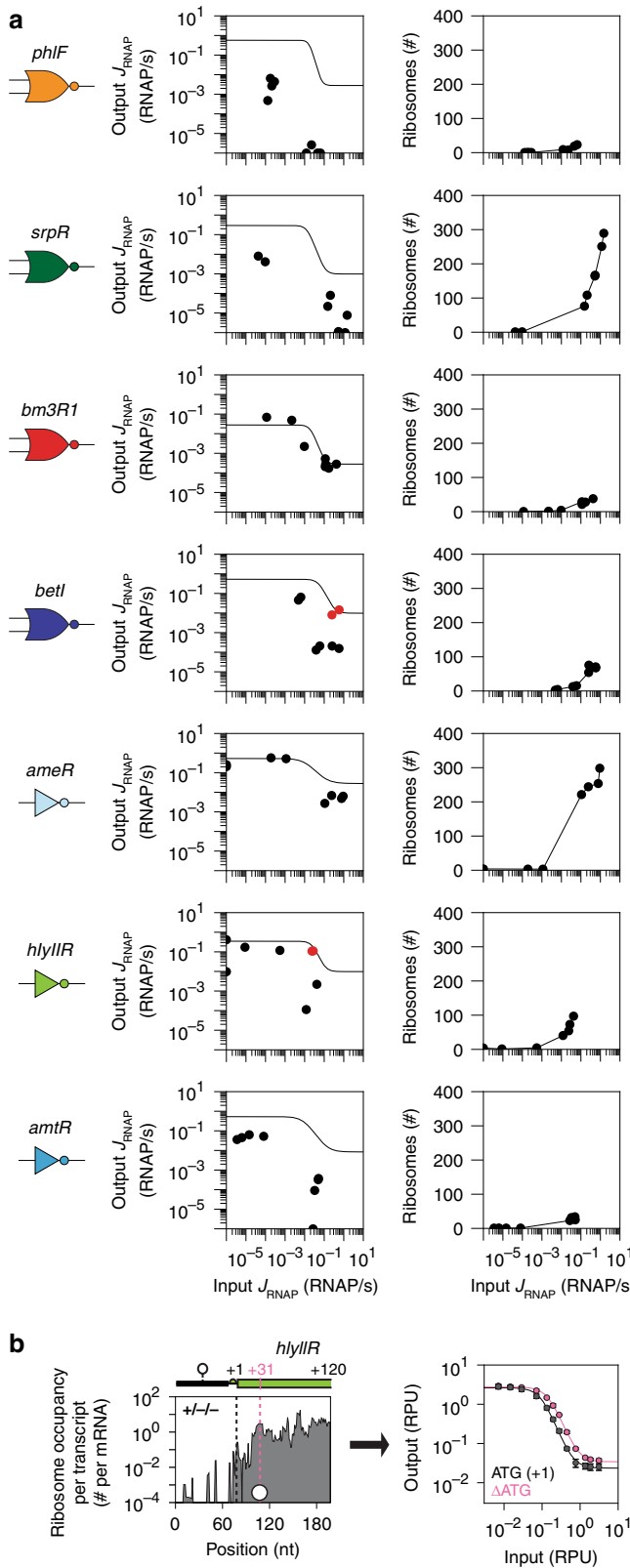

**Fig. 4 Gate responses. a** Data for each gate are shown. (left column) The line indicates the response function, measured when the gate is in isolation[14] (converted to RNAP/s, Methods). The data points correspond to the input and output $J_{RNAP}$ extracted from the RNA-seq data for all eight states. Red dots show states of BetI and HlyIIR gates that have failed. (right column) The number of ribosomes used by the gate as a function of the input promoter activity. **b** Translation from an alternative start codon for *hlyIIR* is shown. Black dashed line is the position of annotated start codon, and white circle (legend in Fig. 2a) and pink dashed line show the position of alternative start codon. The response functions of the original (ATG at +1) and mutated (ATG at +1 deleted) *hlyIIR* are compared. Data are the average ±standard deviations from three independent measurements on separate days. Source data are provided as a Source Data file.

shared ribosome and RNAP usage between gates[25,92,93], although it does not appear to impact all of the gates equally.

The HlyIIR gate is noteworthy because it fails in nearly every way. Its response is weak and, even grossly, it produces the incorrect logic in some states (e.g., +/−/− should be on and −/+/+ should be off) (Figs. 1d, 3a, and 4a). The individual parts show evidence for failures (Fig. 2a). Its RBS is the only one to not have an elevated ribosome occupancy, and the terminator TTS is not in the poly-T region in most states. There is also a spike in ribosome occupancy of *hlyIIR* gene that occurs at a second in-frame ATG internal to *hlyIIR* that includes a predicted upstream RBS (Fig. 4b and Supplementary Fig. 12). Indeed, when the ATG at +1 is deleted, this has little impact on the response function of the HlyIIR gate (Fig. 4b). Finally, for all the repressors we looked for evidence of their expression impacting host genes. Only the *hlyIIR* repressor was found to impact host genes, notably *aceBAK* (glyoxylate cycle), *tonB* (iron uptake), and *mipA* (scaffolding protein) (Supplementary Fig. 17). Sequence analysis of these operons showed putative operators that resemble *hlyIIR* operator sequence (Supplementary Fig. 17). It is remarkable that, despite the complete gate failure, serendipitously this does not impact the circuit function.

**Modeling the circuit robustness to parameter variation**. We were intrigued that the four failed states of these two gates happen to not impact the performance of the overall circuit function. We first performed simple Boolean logic calculations showing the propagation of the states of the inputs through the network (Supplementary Fig. 18). This is a simple treatment, where the inputs are modeled as 0 or 1 and then each NOR and NOT gate responds digitally without additional parameterization. Using this framework, we modeled the experimentally observed failed states and propagated the effect to the final circuit state. We find that the particular circuit topology we use here is robust to all of these changes and the circuit logic that it is designed for is as expected. This might be interpreted as lucky, but it is important to note that we selected this circuit from a set of 60, of which only 45 worked as designed. One of the reasons that the remaining circuits failed may be that they were not similarly robust to the underlying part failures.

Next, we used a more detailed kinetic modeling to understand the circuit dynamics and evaluate the robustness to changes in the parameters. Typically, the problem with such a model is parameterization, where many of the circuit parameters have to be identified from the literature or estimated. In our case, we can obtain nearly all of the parameters for the parts through the RNA-seq and ribosome profiling experiments (Tables 1 and 2, and Supplementary Fig. 19). The time-dependent mRNA and protein levels for each gene was modeled using a set of ordinary differential equations (ODEs) that have parameters corresponding to the strengths of promoters, ribozymes, RBSs, and terminators as

functions are shown as lines in Fig. 4a. Using the RNA-seq data, this can be compared to the gate responses for the eight circuit states. The response functions are surprisingly similar when measured independently versus in the context of the circuit. Two exceptions are the BetI and HlyIIR gates (Fig. 4a). The RNAP fluxes from the gates in the context of the circuit are generally lower than when measured in isolation. This could be due to the

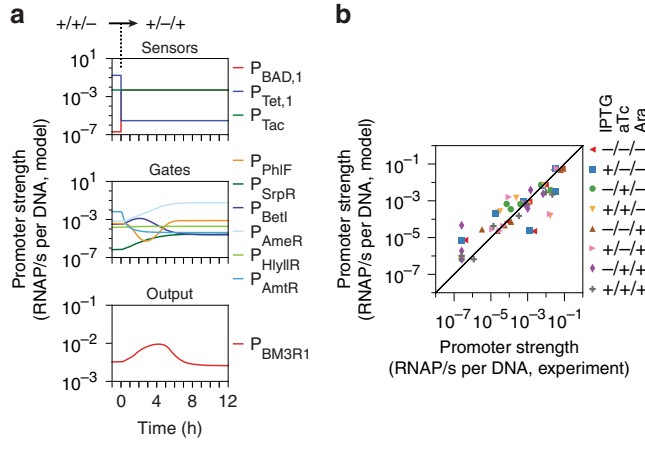

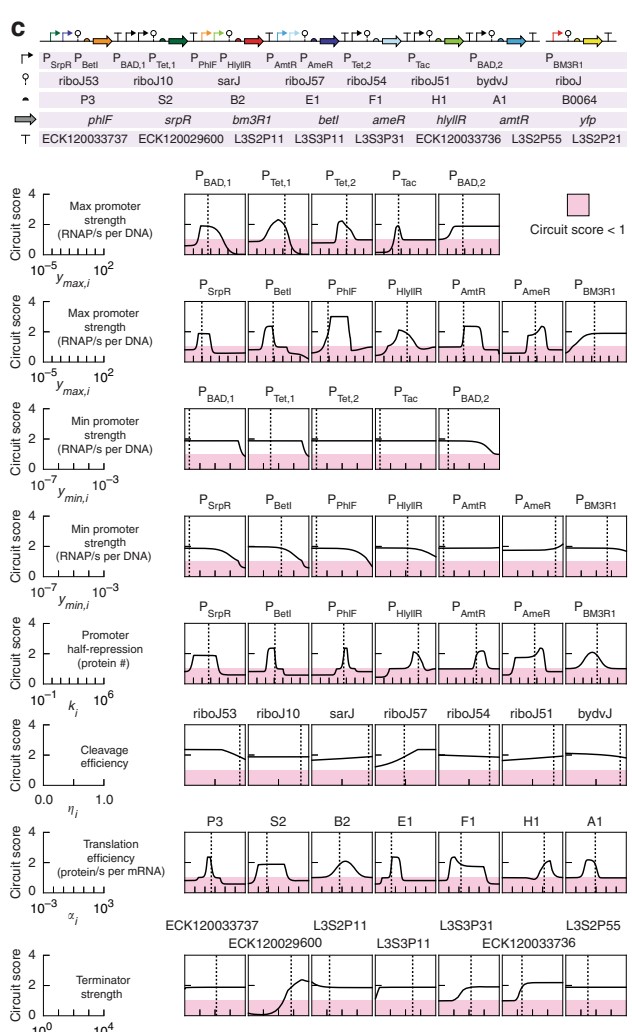

**Fig. 5 Dynamic modelling of genetic circuit using –omics extracted parameters. a** An example of the dynamics predicted when the circuit changes state. The differential equations used for the simulations are provided in the Methods. At $t = 0$, the inputs are switched from $+/+/-$ to $+/-/+$ (IPTG/aTc/Ara) and the response is shown. **b** The predicted steady-state promoter activities of all seven gates in the circuit are compared against their measured activities across all eight induction states of the circuit ($R^2 = 0.73$, $p$ value $= 5.5 \times 10^{-18}$). A line is drawn at x = y to guide the eye. **c** Sensitivity analysis. A circuit diagram showing all genetic parts used in the circuit is shown at the top. When each parameter is varied, all of the remaining parameters are held constant (Methods). Regions in pink show the parameter range that lead to circuit that no longer performs the intended logic operation (circuit score <1). The parameters of the circuit extracted from –omics data are shown as dashed lines (Tables 1 and 2). Source data are provided as a Source Data file.

match all the circuit states, including all of the internal sensor/gate output promoters (Fig. 5b and Supplementary Fig. 20). This demonstrates that the parameters extracted from –omics experiments can be used in conjunction with ODE modeling to predict system performance. Note, however, that discrepancies in the circuit prediction (Fig. 5b) cause the circuit score predicted by the kinetic simulations (2) to be much lower than the values predicted by Cello (80) or measured experimentally (21).

A sensitivity analysis was conducted for each genetic parameter to identify the operational boundaries in which the circuit remains functional, inspired by earlier work on the parameter robustness of natural regulatory networks (Fig. 5c)[95–97]. The circuit score was defined as the lowest expected on state, divided by the highest expected off state in the simulation (Methods). When this score is <1, the circuit is no longer performing the logic operation for which it was designed. Each parameter was varied individually, the eight induction states simulated, and the circuit score calculated. The plots for the robustness of each parameter are shown in Fig. 5c. The starting values are shown as vertical dashed lines for reference.

There are striking differences in the robustness of the circuit design to perturbations in the underlying parts. For example, the $P_{BAD}$ promoter can vary in strength over orders of magnitude with little impact, but the system is very sensitive to the strength of the $P_{Tac}$ promoter. Similarly, the gate promoters show variability in sensitivity, both in the maximum of their response functions and their thresholds. The circuits are insensitive to ribozyme efficiencies, which is not surprising given their primary function is to standardize the mRNAs against errors due to 5′-UTR differences, which are not captured in the model. More surprisingly, there is large robustness to terminator efficiency, where most can be varied over order of magnitude with little impact. The particular terminators that are robust versus sensitive are a function of the circuit topology as well as the particular organization of gates on the DNA sequence.

**The cellular impact of circuit RNAP/ribosome usage.** The RNA-seq and ribosome profiling datasets are then used to calculate the transcriptional and translational costs of holding a gate in a particular state (Fig. 4a). The input to a gate is in units of RNAP/s, representing the usage of transcriptional resources. The number of ribosomes used in each gate can also be calculated by summing all the ribosome occupancies across the gate's repressor gene (Fig. 4a). While the gates have a similar design, they vary in the resources required. For example, the PhlF gate (in our hands, a reliable one) requires low input $J_{RNAP}$ to turn off and <30 ribosomes to operate. In contrast, the SrpR and AmeR gates require high $J_{RNAP}$ to turn off

input parameters (Methods). The only parameters taken from the literature are the mRNA degradation rate and plasmid copy number, which are assumed to be constant, and the only non-omics derived parameter from this work is the growth rate $\mu$. Despite the parameterization being based on static measurements, the equations can be used to predict the circuit dynamics; the response to a change in inputs that leads to a glitch is shown in Fig. 5a[94]. The steady-state solutions are determined and found to

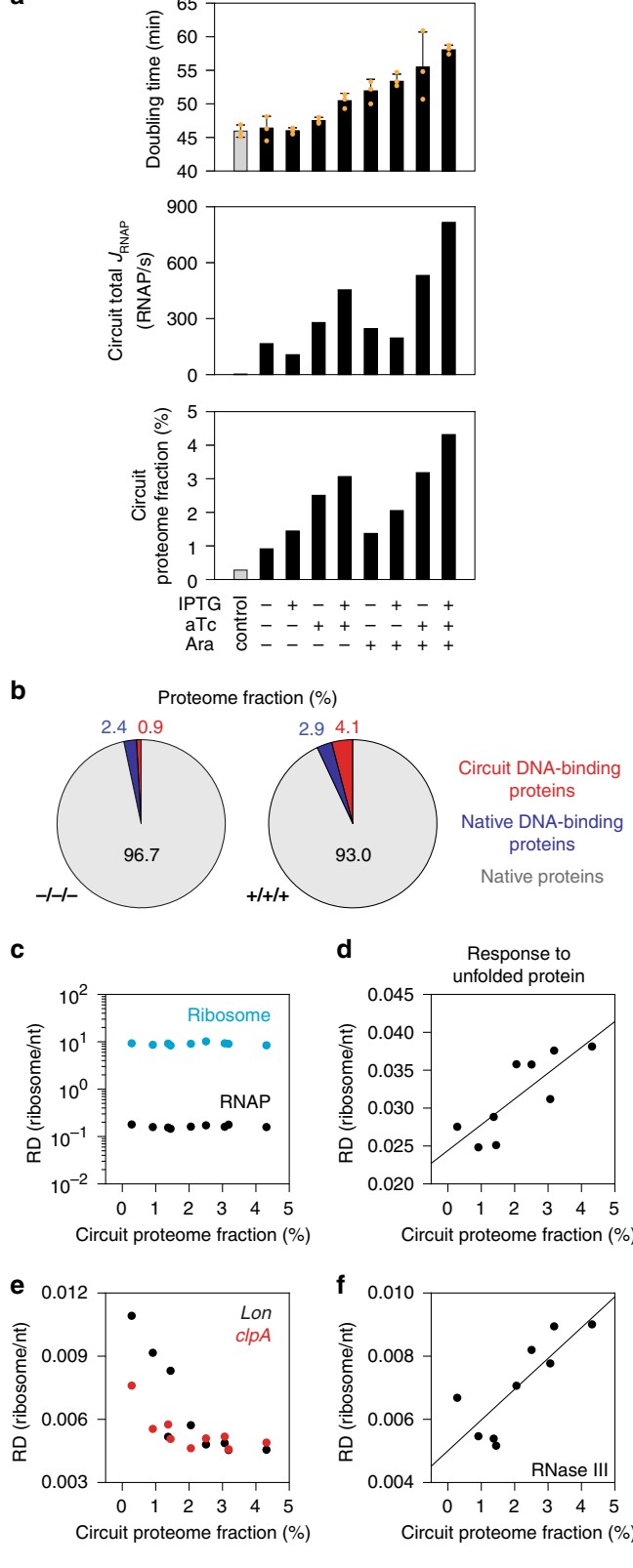

**Fig. 6 Circuit impact on the host cell. a** Comparison of the usage of cellular resources across the eight circuit states. The average cell doubling time (gray and black bars) is calculated from three replicates performed on different days (orange dots) and the error bars represent the standard deviation ($n = 3$). The total RNAP flux is the sum of the activities of all the circuit promoters (each bar is the value from one biological replicate). The proteome fraction is the sum of the estimated fractions for all of the circuit proteins, including the regulators that are part of the sensors and YFP (each bar is the value from one biological replicate). The control is *E. coli* DH10B with pAN871 (circuit backbone, no circuit) and pDV4-P$_{BAD}$-YFP (output backbone with YFP inducible cassette) and includes the expression of the sensor regulatory proteins and YFP (Supplementary Fig. 1). **b** The estimated fraction of the proteome of all 283 native DNA-binding proteins is compared with the 10 regulatory proteins in the synthetic circuit. Native proteins represents all of the remaining genes encoded in the genome. **c** Comparison of the expression of RNAP and ribosome genes for the eight states of the circuit based on the estimated fraction of the proteome the circuit occupies for that state (part **a**). Data for the control are also shown for comparison. The RNAP values represent the sum of the *rpoABCDZ* genes and the ribosome value includes the sum of *rpsABCDEFGHIJKLMNOPQRSTU*, *rplABCDEFIJKLMNOPQRSTUVWXY*, and *rpmABCDEFGHIJ*. **d** Average RD of 12 chaperons involved in unfolded protein response of *E. coli* (*htpG, dnaJ/K, groES/EL, grpE, hslOUV, yegD, ybbN*, and *secB*) is shown. The line is the best fit ($R^2 = 0.66$, *p* value = 0.008). **e** The responses of two housekeeping proteases are shown. **f** The response of RNase III is shown. The line is the best fit ($R^2 = 0.67$, *p* value = 0.007). The nine data points in **d**, **e**, and **f** represent the eight induction states of the circuit as well as the control, as described in **c**. Source data are provided as a Source Data file.

RNAP flux is analogous to the total power usage by the circuit and this varies based on the pattern of transcriptional activity of a state, ranging from 107 to 816 RNAP/s (Figs. 1d and 6a). The estimated fraction of the cellular proteome occupied by circuit proteins can be estimated using the ribosome profiling data (Methods). Depending on the state, up to 4.3% of the proteome consists of regulatory proteins or YFP from the circuit (Fig. 6a). As a comparator, we calculated the estimated fraction of the proteome occupied by the 283 natural DNA-binding proteins in *E. coli* and compared it to the fraction occupied by DNA-binding proteins in the synthetic circuit (Fig. 6b and Supplementary Fig. 21). In state 8 (+/+/+), the circuit produces 40% more DNA-binding proteins than the entire repertoire of DNA-binding proteins in the host cell.

Next, we measured the impact on host gene expression as the circuit consumes more resources. First, we hypothesized that the cell could try to compensate by upregulating RNAP or ribosome expression. We found no upregulation of ribosome or RNAP genes, individually or in combination (Fig. 6c). We then systematically analyzed all chromosomal genes and pathways, looking for positive or negative correlations in their expression (RD) with the estimated fraction of the proteome being used by the circuit (Supplementary Tables 2 and 3) (Methods). There is a downregulation of most of the genes involved in the TCA cycle and aerobic respiration. There is also an upregulation of genes associated with mitigating the impact of the accumulation of heterologous proteins and mRNA. Genes involved in the unfolded protein response, including folding chaperones, are upregulated (Fig. 6d and Supplementary Fig. 22). This has been observed previously[26] and the *htpG* promoter was selected as a means to report burden as a result. For this circuit, we found that *htpG* expression only varies 2-fold and is only weakly correlated with growth rate (Supplementary Fig. 22). Surprisingly, the expression of proteases decreased as the estimated circuit proteome fraction increased (Fig. 6e). RNase III which degrades double stranded RNAs was also overexpressed (Fig. 6f).

and up to 300 ribosomes. A variety of biophysical factors could explain these differences, including the repressor affinities and translational efficiency of transcripts[98,99].

Cells possess a finite pool of cellular resources, and transcription and translation of circuit proteins compete for these resources with the cell's native genes[100–102]. Carrying the circuit slows growth, the magnitude of which depends on the circuit state (Fig. 6a). The total

## Discussion

Computer aided design (CAD) helps the management of large genetic engineering projects that require the selection of many genetic parts to build a system[3,5,14,103–105]. Parts can be selected from databases of previously characterized sequences or designed de novo with computational tools. However, once parts are used in a genetic system, it has been difficult to discern whether they are functioning as intended. Systems are evaluated based on their gross function, such as a circuit function or the production of a small molecule. Tools to interrogate genetic systems have to be able to scale to larger designs, with more parts, without incurring additional cost or experimentation.

Deep sequencing scales appropriately and here we have combined methods that enumerate mRNAs and ribosome usage to fully characterize the parts in a system in a single experiment. While genetic circuits require the evaluation of multiple states, these samples can easily be pooled in a single sequencing lane. Solely using RNA-seq and ribosome profiling requires some parameters to be pulled from other sources, including degradation rates, plasmid copy number, and total number of active ribosomes, all of which are not constant and depend on environmental conditions and the growth rate[29,30,64,106,107]. Beyond these techniques, the rapid decline in deep sequencing costs has led to a suite of techniques that could be performed simultaneously to offer further insight into the workings of an engineered system in a cell[108]. New sequencing methods have been developed for the detection of TSSs, mRNA degradation, RNA structures, RNAP elongation, transcription dynamics, enzyme activity, metabolomics, and translational initiation/elongation/termination[109–123]. The single-cell resolution obtained from fluorescent reporters and flow cytometry and microscopy[124] is currently lost, but this could be recovered as single-cell RNA-seq experiments become more practical and they are able to provide the full transcript profiles, rather than just the mRNA levels of genes[125–127]. Automating these techniques in experimentation and data handling will be a powerful debugging tool.

Here, we chose one of the most well-behaved circuits we have constructed to date. We were shocked at the number of internal failures. Many parts do not function as designed, bring unintended regulatory sequences in their DNA (e.g., cryptic promoters) or introduce new functions due to the new sequences formed when two parts are connected. Remarkably, a gate (HlyIIR) fails entirely. Indeed, there is some luck in our selection of a circuit whose topology is robust to the gate and parameter variations observed experimentally. However, if these same parts are used in a different system, they could cause it to fail, the probability of which increases as the size of the project grows. This is reminiscent of an earlier observation by Leibler and co-workers when they randomly combined regulatory parts and noticed that the circuit often does not behave as can be explained by the known part functions[2]. Similar effects are seen in the combinatorial assembly of metabolic pathways[35,128].

Haphazardly finding that systems work or fail for unknown reasons leads to the mystic of biology being labeled unpredictable when in reality it is only our inability to observe how a design actually works in a cell that inhibits precision engineering. This work focuses on the analysis of a complete circuit where the individual parts were not pre-characterized using–omics tools and the design algorithm did not consider this type of information. One can imagine going in after the fact and mutating problems, like cryptic promoters, to fix them or to replace malfunctioning parts. However, it would be more valuable to run –omics experiments routinely on the subsystems to correct problems prior to their use, particularly when the intention is to reuse them in many permutations (e.g., in the Cello User Constraint File). Gates could also be removed from this list completely, as we have now done for HlyIIR. The careful integrative steps of design and debugging will lead to their more reliable assembly later. In addition, the characterization of the total RNAP and ribosomes used by a gate allows for the design of low power variants that minimize the impact on the cell. This will lead to more reliable CAD in genetic engineering that can predict failure modes and warn a user if there is a potential growth defect or the time expected before there is evolutionary breakage.

## Methods

**Strain, media, and inducers**. The Escherichia coli DH10B derivative NEB 10-beta Δ(ara-leu) 7697 araD139 fhuA ΔlacX74 galK16 galE15 e14- φ80dlacZΔM15 recA1 relA1 endA1 nupG rpsL (StrR) rph spoT1 Δ(mrr-hsdRMS-mcrBC) was used as the host (New England Biolabs, MA, C3019). Cells were grown in M9 minimal media, consisting of 1× M9 media salts (Sigma–Aldrich, MO, M6030), 0.34 g/L thiamine hydrochloride (Sigma–Aldrich, MO, T4625), 0.4% D-glucose (Sigma–Aldrich, MO, G8270), 0.2% Casamino acids (Acros, NJ, AC61204-5000), 2 mM MgSO$_4$ (Sigma–Aldrich, MO, 230391), and 0.1 mM CaCl$_2$ (Sigma–Aldrich, MO, 449709). The inducers used in this study were isopropyl β-D-1-thiogalactopyranoside (IPTG; Sigma–Aldrich, MO, I6758), anhydrotetracycline hydrochloride (aTc; Sigma–Aldrich, MO, 37919), and L-arabinose (Ara; Sigma–Aldrich, MO, A3256). Antibiotic selections were performed with 50 μg/ml kanamycin (Gold Biotechnology, MO, K-120-5) and 50 μg/ml spectinomycin (Gold Biotechnology, MO, S-140-5). Phosphate buffered saline (1x PBS) solution was prepared from a 10x solution purchased from OmniPur (MA, 6505).

**Circuit induction**. E. coli was co-transformed with the pAN3945 (circuit) and pAN4023 (output) plasmids[14] (Supplementary Fig. 1). Individual colonies were picked and inoculated into 5 ml M9 minimal media with kanamycin and spectinomycin in a 14 ml culture tube (Corning, MA, 352059). The culture was grown overnight for 16 h at 37 °C and 250 rpm in an Innova 44 shaker (Eppendorf, CT). The following day, cultures were diluted 23 μl into 4 ml fresh M9 media with antibiotics and grown under the same conditions for 3 h. For induction, cells were diluted a second time 380 μl into 250 ml M9 media with antibiotics in 3 L Erlenmeyer flasks (Pyrex) in the presence or absence of 1 mM IPTG, 2 ng/ml (4.32 nM) aTc, and 5 mM Ara. Culture flasks were incubated at 37 °C and 200 rpm for 5–7 h. At OD$_{600}$ = 0.3, cells were collected, rapidly filtered at room temperature by passing through a 0.22 μm pore size nitrocellulose filter (Sigma–Aldrich, MO, N8645), and carefully scraped from the surface of the filter using a stainless-steel scoopula (Thermo Fisher Scientific, USA, 14-357Q) and flash frozen immediately in liquid nitrogen. Cell pellets were saved in −80 °C until use in RNA-seq and ribosome profiling steps.

**Fluorescence measurement**. Individual colonies of wild-type E. coli as well as E. coli cells harboring circuit plasmids (pAN3945 and pAN4023) and E. coli cells harboring a reference plasmid (pJSBS.RPU) (Supplementary Fig. 1) were picked and inoculated into 200 μl M9 minimal media with no antibiotic (for wild-type cells), with kanamycin and spectinomycin (for circuit cells), and with kanamycin (for reference cells) and grown overnight at 37 °C and 1000 rpm in an ELMI Digital Thermos Microplates shaker incubator (ELMI Ltd, Latvia, hereby ELMI plate shaker) and using Nunc$^{TM}$ 96-well plates (Thermo Scientific, USA, 249662). Next day, cells were diluted 178-fold (two serial dilutions of 15 μl into 185 μl) into prewarmed M9 media with proper antibiotics, and were incubated for 3 h at 37 °C and 1000 rpm in ELMI plate shaker using Nunc$^{TM}$ 96-well plates. Cells were then serially diluted 658-fold in two steps. First, 15 μl cells were added into 185 μl prewarmed M9 media with proper antibiotics and inducers (only circuit cells were induced with presence or absence of 1 mM IPTG, 2 ng/ml (4.32 nM) aTc, and 5 mM Ara). Next, 20.3 μl of mixture was added into 980 μl prewarmed M9 media with proper antibiotics and inducers in a PlateOne® Deep 96-well plate (USA scientific, USA, 1896–2000), and culture was incubated for 5 h at 37 °C and 900 rpm in a Multitron Pro incubator shaker (In Vitro Technologies, VIC, Australia). Finally, after 5 h induction, 20 μl of culture was added to 180 μl 1× PBS solution with 2 mg/ml kanamycin to stop translation and cell growth, and the mixture was incubated for one hour before fluorescence was measured using flow cytometry.

**Flow-cytometry analysis**. Fluorescence was measured using an LSRII Fortessa flow cytometer (BD Biosciences, San Jose, CA) using BD FACSDiva software, version 8.0.3. More than 10,000 gated events were collected for each sample. The flow cytometer software FlowJo, version 9 (TreeStar, Inc., Ashland, OR) was used to gate the raw data and calculate the median YFP fluorescence values for the gated events (Supplementary Fig. 23).

**Conversion to relative promoter units (RPU)**. The activity of a circuit's output promoter (P$_{BM3R1}$) can be calculated in RPU by comparing circuit's fluorescence to that of the plasmid (pJSBS.RPU, Supplementary Fig. 1) containing a reference promoter (P$_{J23101}$); RPU = (<YFP> - <YFP>$_0$)/(<YFP>$_{RPU}$ - <YFP>$_0$), where <YFP> is the median fluorescence of circuit at each induction state, <YFP>$_{RPU}$ is

the median fluorescence of cell containing the reference plasmid, and $<YFP>_0$ is the median autofluorescence of wild-type *E. coli* DH10B.

**Cello predictions**. Cello version 1.0 was used with the Eco1C1G1T1 UCF (http://cellocad.org/). The sensor activities used were (on and off): $P_{Tac} = 2.8$ and $0.0034$ RPU, $P_{BAD} = 2.5$ and $0.0082$ RPU, and $P_{Tet} = 4.4$ and $0.0013$ RPU. To calculate promoter activities, the input and output promoters of all NOT and NOR gates in the circuit are identified (Fig. 1b). The output promoter activity $y$ for each gate (in RPU) is calculated according to

$$y = y_{min} + (y_{max} - y_{min})\frac{K^n}{K^n + x^n} \tag{1}$$

where $x$ is the input promoter activity (in RPU), and $y_{min}$, $y_{max}$, $K$, and $n$ are constants for each gate. For the NOR gates, the input promoter activity $x$ is the sum of the activities of the two input promoters $x = x_1 + x_2$. The activity of $P_{BM3R1}$ output promoter is multiplied by 0.44 to account for lower copy number of the plasmid carrying output reporter *yfp*. All Cello predicted promoter activities are in RPU, but can be converted to absolute units using conversion 1 RPU = 0.019 RNAP/s per promoter (see below).

**RNA-seq library preparation and sequencing**. RNA-seq sample preparation was carried out following the method described earlier[54,55]. Briefly frozen cell pellets from circuit induction step were mixed with 650 μl frozen droplets of lysis buffer (20 mM Tris (pH 8.0), 100 mM $NH_4Cl$, 10 mM $MgCl_2$, 0.4% Triton X-100, 0.1% Tergitol, 1 mM chloramphenicol and 100 U/ml DNase I) and added to a 25 ml canister (Retsch, Germany, 014620213) already prechilled in liquid nitrogen, and pulverized for 15 min in five cycles of 3 min with intermittent cooling between cycles using a TissueLyser II (Qiagen) set at 15 Hz. The pulverized cell pellets were quickly transferred to prechilled microcentrifuge tubes and spun down at $20,000 \times g$ for 10 min at 4 °C using a refrigerated benchtop centrifuge (Eppendorf, CT) and supernatant was collected as the cell lysate. This cell lysate was split to be used separately for RNA-seq and ribosome profiling preparation steps. The following steps were performed on the RNA-seq portion only. The total RNA was extracted from the split lysate using previously described hot phenol–SDS extraction method[129]. Ribosomal RNAs were removed from 20 μg of total RNA using the MICROBExpress kit (Thermo Fisher Scientific, USA, AM1905), followed by fragmentation of the remaining mRNAs, tRNAs, and 5 S rRNAs using RNA fragmentation reagents (Invitrogen, USA) at 95 °C for 90 s. The RNA fragments (10–45 nt) were selected and recovered from a 15% TBE–urea denaturing poly-acrylamide gel (Thermo Fisher Scientific, USA, EC6885BOX). Briefly, the correct band was excised from the gel and transferred into a 0.5 ml tube, where the bottom of the tube was poked through. The tube was then placed in a 2 ml screw cap tube and spun down at $20,000 \times g$ for 3 min. The collected gel pieces were incubated with 0.5 ml of 10 mM Tris 7.0 for 10 min at 70 °C in a shaking thermomixer. After that the mixture was transferred to a Spin-X cellulose acetate column (Thermo Fisher Scientific, USA, 07-200-388) and spun down at $20,000 \times g$ for 3 min. The solution was then transferred to a new tube and precipitated by adding 55 μl of 3 M NaOAc (pH 5.5), 2 μl of glycoblue, and 550 μl of 100% isopropanol, and chilled at −80 °C for 30 min. RNA was then pelleted by spinning down the mixture at $20,000 \times g$ for 30 min at 4 °C, followed by aspirating the supernatant, washing the RNA pellet one more time with 750 μl of 80% ethanol at 4 °C, air drying it for 5 min, and resuspending it in 15 μl of 10 mM Tris 7.0 for cDNA preparation. Next, cDNA libraries were generated by reverse-transcribing the purified RNA fragments using Superscript III (Invitrogen, USA, 18080044) with the oCJ485 primer (/5Phos/AGATCGGAAGAGCGTCGTGTAGGGAAAGAGTGT/iSp18/CAAGCAGAAGACGGCATACGAGATATTGATGGTGCCTACAG) at 50 °C for 30 min. Original RNA was hydrolyzed and removed from the mixture by adding 0.1 M NaOH, followed by incubation at 95 °C for 15 min. The expected cDNA bands (125–150 nt) were selected and recovered from a 10% TBE–urea polyacrylamide gel (Thermo Fisher Scientific, USA, EC6875) following the steps described above, except that Tris 8.0 was used instead of Tris 7.0, and a mixture of 32 μl of 5 M NaCl and 1 μl of 0.5 M EDTA was used instead of 55 μl of 3 M NaOAc (pH 5.5). The cDNA products were circularized by adding them to a 20 μl reaction volume supplemented with 2 μl of 10× CircLigase buffer, 1 μl of 1 mM ATP, 1 μl of 50 mM $MnCl_2$, and 1 μl of CircLigase (Epicenter, USA). The reaction mixture was incubated at 60 °C for 2 h followed by heat inactivation at 80 °C for 10 min. 5 μl of the circularized DNA was amplified using Phusion HF DNA polymerase (New England Biolabs) with the o231 primer (CAAGCAGAAGACGGCATACGA) and indexing primers (AATGATACGGCGACCACCGAGATCTACACGATCGGAAGAGCACACGTCTGAACTCCAGTCACNNNNNNACACTCTTTCCCTACAC) for 7–10 cycles. The amplified products were size selected using an 8% TBE–urea polyacrylamide gel (Thermo Fisher Scientific, USA, EC62152) and libraries between 125 and 150 nt were recovered from gel as described above. These final products were analyzed using a BioAnalyzer (Agilent, USA), and were pooled together with those from ribosome profiling steps into three sequencing runs (each sample with a unique indexing primer). Sequencings were performed on an Illumina HiSeq 2500 in rapid-run mode using sequencing primer CGACAGGTTCAGAGTTCTACAGTCCGACGATC. To process raw reads acquired from RNA-seq runs, we first trimmed the adaptors (CTGTAGGCACCATCAATATCTCGTATGCCGTCTTCTGCTTG) using Cutadapt[130], where we only selected trimmed reads

longer than 10 nt. Next, reference sequences, in FASTA format, were generated for each sample, where they include DNA sequences of both the *E. coli* DH10B genome (NC_010473.1) and the circuit and reporter plasmids (pAN3945 and pAN4023 for cells containing circuit, and pAN871 and pDV4-$P_{BAD}$-yfp for control cells with empty plasmid backbones). These reference sequences were first indexed using the 'bowtie-build' function of Bowtie 1.1.2 sequence alignment[131], followed by aligning all trimmed reads to these indexed references with the 'bowtie' function with parameters '-v1 -m2 -k1'. All information about aligned reads such as their reference (genome or plasmid), position (5′ and 3′), strand (forward or reverse), and mapping quality are stored in an output file to be used for transcription profile generation.

**Construction of transcript profiles**. Mapped reads at each nucleotide position were normalized by the total number of mapped nucleotides in the sample and then multiplied by a $10^9$ factor. The tRNAs in our samples comprised between 15 and 65% of the transcriptome across different induction states of the circuit. Although end-enriching method can visualize the exact tRNA processing sites at both 5′- and 3′-ends of tRNAs (Supplementary Fig. 4a), the non-uniform fraction of tRNAs between our samples (Supplementary Fig. 4b) can introduce bias against other regions in the circuit and genome, and can distort our genetic parts characterizations. To eliminate the bias, we manually removed all reads mapped to the tRNA regions before calculating the total number of mapped nucleotides used for normalizing RNA-seq profiles. FPKM (fragments per kilobase of transcript per million mapped reads) of each gene was then calculated by averaging the height of RNA-seq profile over the length of the gene. In addition, at steady-state, RNAP flux at nucleotide position x can be calculated from $J(x) = \gamma M(x)$, where $M(x)$ is the height of the normalized RNA-seq profile (or mRNA transcript levels) and $\gamma = 0.0067\ s^{-1}$ is the mRNA degradation rate[8,63]. The resulting RNAP flux $J(x)$ has arbitrary units.

**Conversion of RPU to RNAP/s**. The RNAP flux can be reported as either the total for all the copies of a promoter in the cell (e.g., carried on a multi-copy plasmid) or per promoter, where it is normalized by the copy number. Single-molecule measurements yielded a conversion factor of 1 RPU = 0.019 RNAP/s per promoter for the BBa_J23101 reference promoter[17,18] in the same plasmid backbone we use in this manuscript (pJSBS.RPU, Supplementary Fig. 1). The average copy number of this plasmid was measured to be 9, so the total flux from the promoter is 0.171 RNAP/s. The reference promoter is not in the genetic circuit, so it is not used for the RNA-seq experiments and we need a different method to connect the promoter strengths to RPU. To do this, we use the circuit's output promoter ($P_{BM3R1}$) because we know its strength in RPU from the flow-cytometry experiments (Fig. 1c) and it is present in the circuit, so we can measure the promoter strength as a change in RNAP flux $\delta J$ in arbitrary units. Considering all circuit states, the eight $\delta J$ from RNA-seq are plotted against the RPU from cytometry to obtain a conversion factor ($R^2 = 0.87$, $p$ value = 0.0008, Supplementary Fig. 5). The circuit and reference plasmids are based on the same p15a backbone, but the output plasmid is based on a pSC101 backbone, which has a 2.25-fold lower copy number[18]. The conversion of the height of the transcript profile $M(x)$ in arbitrary units to that of RNAP flux $J(x)$ in the unit of RNAP/s per promoter is then: $J(x) = 2.25 \times 0.019 \times 5.05 \times 10^{-5}\ [\gamma M(x)]^{1.64} = 2.16 \times 10^{-6}\ [\gamma M(x)]^{1.64}$. Finally, to calculate the total RNAP flux for all copies of a promoter, $J(x)$ is multiplied by 9. The RNAP flux profile of the circuit at all induction states is shown in Supplementary Fig. 24 for both sense and antisense strand of the circuit DNA sequence.

**Ribosome profiling**. As described above, the cell lysate was split to be used for both RNA-seq and ribosome profiling steps. The following steps are performed on the portion saved for ribosome profiling. The total RNA was extracted from the lysate using previously described hot phenol–SDS extraction method[129]. The RNA was diluted to 0.5 mg in 200 μl of the lysis buffer including 5 mM $CaCl_2$ and 100 U SUPERase·In (Invitrogen, USA) and was digested with 750 U micrococcal nuclease S7 (Sigma–Aldrich, MO, 10107921001) at 25 °C for 1 h to perform ribosome footprinting, followed by quenching with 6 mM EGTA (Bioworld, TX, 40520008-2). The footprinting products were kept on ice before they were spun in a sucrose-density-gradient (10 – 55% w/v) using an ultracentrifuge (Beckman Coulter, Atlanta, GA) with a SW41 rotor at 35,000 rpm ($151,194 \times g$) for 165 min, and the monosome fraction was collected by a gradient fractionator (BioComp Instruments, New Brunswick, Canada). To collect ribosome-protected mRNA fragments, hot phenol–SDS extraction was performed[129]. Next, mRNA fragments 10–45 nucleotides in length were selected and recovered from a 15% TBE–urea denaturing polyacrylamide gel (Thermo Fisher, MA, EC6885BOX) as described above. The cDNA libraries were generated by Superscript III and were subsequently circularized using CircLigase as described in RNA-seq preparation above. Fragments belonging to ribosomal RNA were removed by a biotinylated oligo mix for *E. coli*. Similar to RNA-seq, 5 μl of circularized DNA was amplified for 7–10 cycles using Phusion HF DNA polymerase and with the o231 primer and the sample-specific indexing primer. Only amplified products between 125 and 150 nt were selected and recovered from an 8% TBE–urea polyacrylamide gel as described above. Finally, these purified cDNA libraries were analyzed with a BioAnalyzer, and pooled as described above, and sequenced on an Illumina HiSeq 2500 in

rapid-run mode using sequencing primer CGACAGGTTCAGAGTTCTACAGT CCGACGATC. Similar to RNA-seq data processing steps, adaptors (CTGTAGGC ACCATCAATATCTCGTATGCCGTCTTCTGCTTG) were first trimmed off the raw reads acquired from ribosome profiling runs using Cutadapt[130] (read length > 10 nt). These trimmed reads were then aligned to the same indexed reference sequences generated for RNA-seq. Alignments were performed using the 'bowtie' function of Bowtie 1.1.2[131] with parameters '-v1 -m2 -k1'. The output files stored all necessary information about the aligned reads (reference, position, strand, and mapping quality) to be used for the ribosome occupancy profile generation.

**Construction of ribosome occupancy profile.** A center-weighting approach was taken to map the P-site position of ribosomes on all aligned footprint reads ranging from 23 to 42 nucleotides in length. For each aligned footprint read, 11 nucleotides on either end were removed and the nucleotides in the remaining region were given the same score, normalized by the length of this center region[54]. Therefore, the ribosome occupancy at each nucleotide position is the sum of all the center-weighted scores from all the reads aligned to that nucleotide. The ribosome occupancy at each nucleotide was then normalized by the sum of all ribosome occupancies across DNA sequences in that sample (including both genome and circuit), and then multiplied by the total number of active ribosomes in *E. coli* which is a function of cell growth rate (20,000 at 45 min doubling time[64]). The resulting ribosome occupancy profile has an absolute unit which is the number of ribosomes at each position. Contrary to RNA-seq, applying tRNA correction was not necessary for ribosome occupancy profiles since ribosomes do not bind tRNAs and therefore tRNA contamination is negligible. We also generated a profile of ribosome occupancy per transcript by dividing the ribosome occupancy at each nucleotide position by the amount of RNAP flux (in unit of RNAP/s) at the same position, multiplied by mRNA degradation rate ($\gamma = 0.0067\ \text{s}^{-1}$)[8,63]. The resulting profile has absolute units (number of ribosomes per mRNA) and is shown in Supplementary Fig. 25 for all states of the circuit on both sense and antisense strands. The ribosome density (RD) of each gene was calculated by averaging all ribosome occupancies over the length of the gene, in absolute units. However, three additional normalizations were needed to calculate RDs (following approaches by Li and Weissman[54,55]): (1) to remove the effects of translation initiation and termination on ribosome density, the first and last five codons of each gene were excluded from averaging calculation, (2) to account for the elevation of ribosome occupancies at the beginning of the genes, an exponential decay function was fitted to a genome-wide occupancy profile and this fitted function was used to normalize the ribosome occupancy at each nucleotide of a given gene, and (3) to reduce the effect of outlier ribosome occupancies, the top and bottom 5% of the ribosome occupancies over the length of the gene were removed (90% winsorization) if the average read density on a gene was higher than one.

**TSS/TTS identification.** Transcriptional start sites (TSSs) across DNA sequences are identified by calculating the ratio of RNAP flux $J_{RNAP}$ between each neighboring positions, $J_{RNAP}(x + 1)/J_{RNAP}(x)$. For TSSs in the reverse strand, the inverse of the above ratio is calculated. We also applied a threshold of ratio >5 to call a TSS. As a limitation to our method, for promoters located immediately upstream of a ribozyme, the identified TSS is the same as ribozyme cleavage site, since the cleaved RNA product is ~10 nt long and is lost during RNA-seq sample preparation, creating a sharp increase in RNAP flux profile at the cleavage site.

Transcriptional termination sites (TTSs) are identified by calculating an averaged-window ratio $AWR(x)$ profile for each nucleotide position $x$

$$AWR(x) = \frac{\frac{1}{n}\left[\sum_{i=x-n}^{x} J(i)\right]}{\frac{1}{n}\left[\sum_{i=x}^{x+n} J(i)\right]} \quad (2)$$

where $n = 10$ is the length of the averaging window. For reverse strand, $AWR(x)$ is the inverse of Eq. 2. Similar to promoters, we applied a threshold $AWR(x) > 5$. Because of the gradual decrease in RNAP flux at the termination sites, a continuous span of multiple nucleotides can pass this threshold. To identify the most probable termination site within this span, position x with maximum $AWR(x)$ value is marked as transcriptional termination site TTS.

**Characterization of the Δ(ATG) HlyIIR gate.** The response function was characterized as previously described[14]. A version of the pHlyIIR-H1 plasmid[14] was constructed (pHlyIIR-H1-ΔATG, Supplementary Fig. 1 and Supplementary Table 4) where Gibson assembly was used to delete the start codon Δ(ATG) of *hlyIIR*. The original and mutated *hlyIIR* plasmids were transformed into *E. coli* DH10B. In addition, a pAN1818 plasmid[14] carrying an IPTG-inducible promoter P_Tac for *yfp* transcription (Supplementary Fig. 1) was also used to transform *E. coli* DH10B. Single colonies of these three strains as well as an *E. coli* strain carrying the reference RPU plasmid (pJSBS.RPU, Supplementary Fig. 1) were inoculated into 200 μl of M9 media with kanamycin in Nunc™ 96-well plates and grown for 16 h at 37 °C and 1000 rpm in ELMI plate shaker. On the next day, cells were diluted 178-fold (two serial dilutions of 15 μl into 185 μl) into M9 media with kanamycin, and were incubated for 3 h at 37 °C and 1000 rpm in an ELMI plate shaker using Nunc™ 96-well plates. Cells were then diluted 658-fold (two serial dilutions of 15 μl into 185 μl, and then 3 μl into 145 μl) into M9 media with kanamycin, and were induced with varying concentrations of IPTG (0, 5, 10, 20, 30,

40, 50, 70, 100, 150, 200, and 1000 μM). After induction, cells were grown in Nunc™ 96-well plates for 5 h at 37 °C and 1000 rpm in an ELMI plate shaker, and then 40 μl of culture was added to 160 μl of 1× PBS solution with 2 mg/ml kanamycin. This mixture was incubated for 1 h and fluorescence was measured using flow cytometry and converted to RPU (above). At each IPTG concentration, the measured RPU of pAN1818 plasmid represents the input promoter activity x to the both original and mutated HlyIIR gates, while the measured RPU of either *hlyIIR* plasmids is the output promoter activity y of the gates. The output activity y of each gate was plotted as a function of the input activity x for all IPTG inductions, and a Hill function was fit to data according to Eq. 1. Fitting was performed in Python with minimize function of scipy using the SLSQP method.

**Promoter strength calculation.** To characterize promoter strength δJ, first the TSS of each promoter was identified (above). Using the identified TSS, the promoter activity is calculated as

$$\delta J = \frac{1}{n}\left[\sum_{i=X_{TSS}+A}^{X_{TSS}+A+n} J(i) - \sum_{i=X_{TSS}-A-n}^{X_{TSS}-A} J(i)\right] \quad (3)$$

where $J(i)$ is RNAP flux at position i, $n = 10$ is the length of averaging window, and A = 10 is the gap between TSS and averaging window. For promoters in reverse strand, the δJ value calculated from Eq. 3 is multiplied by −1.

**Calculation of ribozyme activity.** The cleavage site of each ribozyme was identified as a result of TSS calculation (above) for the promoter located immediately upstream of each ribozyme. Next, for each ribozyme, all the cut and uncut fragments around the cleavage site were identified. These are the raw RNA-seq fragments used to generate transcription profiles with. Cut fragments are those that either end exactly at the cleavage site or begin with it. Uncut fragments are those that overlap the cleavage site (Supplementary Fig. 13). The total number of cut fragments downstream of the cleavage site $F_{cut}$ was quantified as well as the total number of uncut fragments $F_{uncut}$. Ribozyme cleavage efficiency is calculated as $CE = F_{cut}/(F_{cut} + F_{uncut})$.

**Calculation of RBS strength.** The translation efficiency (TE) of an RBS was calculated using the ribosome density (RD) and the steady-state mRNA expression level of the gene associated with the RBS. The number of mRNA transcripts at steady-state ($m_{SS}$) is calculated by averaging RNAP flux profile (in unit of RNAP/s) over a 10 nt window at the end of each gene (−10 to 0), divided by mRNA degradation rate ($0.0067\ \text{s}^{-1}$)[8,63]. Note that for *amtR* gene, the averaging window was shifted 150 nucleotides into the coding section (−160 to −150) to avoid the RNAP flux by an internal cryptic promoter. Next, translation efficiency is calculated as $TE = (RD \times \omega)/m_{SS}$, where $\omega = 15\ \text{s}^{-1}$ is the ribosome elongation rate[107]. TE represents the number of ribosomes that finish translation and produce a new molecule of protein per mRNA transcript per second.

**RBS Calculator and mRNA secondary structure prediction.** The translation initiation rate (TIR) for each RBS sequence was predicted using RBS Calculator (v2.0) (https://salislab.net/software/predict_rbs_calculator)[90,91]. The total Gibbs free energy change $\Delta G_{total}$ is calculated and used to determine $TIR = Ke^{-\beta \Delta Gtotal}$, where $\beta = 0.45\ \text{mol/kcal}$ is the Boltzmann constant, $K = 2500$ is a constant, and $\Delta G_{total}$ is the total energy difference between ribosome-free and ribosome-bound mRNA. For all calculations, 35 nucleotides before and after start codon was used as mRNA sequence and *E. coli*'s 16 S rRNA (5′-ACCUCCUUA-3′) was used as anti-Shine-Dalgarno sequence.

RNA folding Gibbs free energy (ΔG) and RNA secondary structure of ribozyme insulators were calculated using ViennaRNA software (version 1.8.5, default options)[132]. For ΔG calculations, the full sequence of each ribozyme was used. RNA secondary structures were also predicted for two ribozymes (sarJ and riboJ57) where their transcription started from an upstream promoter adding 63 extra nucleotides to the 5′-UTR sequence (Supplementary Fig. 14).

**Calculation of mRNA abundance.** The number of full length mRNA molecules for each gene at steady-state ($m_{SS}$) is calculated as the ratio of mRNA transcription rate (in unit of RNAP/s) and mRNA degradation rate (above).

**Calculation of terminator strength.** First, the TTS was identified following the averaged-window ratio (AWR) method described above. $T_S$ is defined as the fold-decrease in RNAP flux before and after a TTS as

$$T_S = \frac{\frac{1}{n}\left[\sum_{i=X_{TTS}-A-n}^{X_{TTS}-A} J(i)\right]}{\frac{1}{n}\left[\sum_{i=X_{TTS}+A}^{X_{TTS}+A+n} J(i)\right]} \quad (4)$$

where $J(i)$ is the RNAP flux at position $i$, $n = 10$ is the length of averaging window, and A = 10 is the gap between TTS and averaging window. For terminators in reverse strand, the $T_S$ value calculated from Eq. 4 is inverted.

**Calculation of ribosome usage**. The ribosome usage is defined by the number of ribosomes sequestered by a gene at steady state, and is calculated by summing the height of ribosome occupancy profile over the length of the gene.

**Calculation of proteome fraction**. Ribosome profiling data are used to estimate the fraction of total cellular proteome $\Phi(i)$ that is occupied by a particular gene product $i$ as

$$\Phi(i) = \frac{\text{RD}_i \text{MW}_i}{\sum_k \text{RD}_k \text{MW}_k} \tag{5}$$

where $\text{RD}_i$ is ribosome density of gene $i$ and $\text{MW}_i$ is the molecular weight of its protein product[54]. The estimated proteome fraction of the circuit at each induction state is calculated by summing the proteome fraction of its regulatory proteins (repressors and sensors) and *yfp* reporter. This value was compared against the estimated total proteome fraction of all DNA-binding regulatory proteins in *E. coli* genome. A list of 302 unique DNA-binding proteins in *E. coli* MG1655 was identified using EcoCyc database (https://ecocyc.org/)[133], from which 283 were found to be present in *E. coli* DH10B. In addition, the impact of estimated circuit proteome fraction on cellular metabolism (below), RNAP and ribosome expression, and cellular maintenance for proteins and mRNAs was studied. To calculate the RD of ribosomes, the RD of all 54 ribosomal proteins in *E. coli* DH10B (*rpsABCDEFGHIJKLMNOPQRSTU* in 30 S and *rplABCDEFIJKLMNOPQR-STUVWXY* and *rpmABCDEFGHIJ* in 50 S subunits) were summed as an estimate for the RD of ribosomes. Similarly, RD of RNAP was estimated as the sum of the RD of *rpoBCZ*, RD of *rpoD* (σ70), and half of the RD of *rpoA* due to its 2:1 stoichiometry in RNAP. Finally, the average RD of genes involved in the unfolded protein response of *E. coli* (*htpG, dnaJ/K, groES/EL, grpE, hslOUV, yegD, ybbN, secB*, taken from EcoCyc[133]) was calculated and correlated with the change in the estimated circuit proteome fraction across induction states.

**EdgeR and gene ontology analysis**. To study the impact of estimated circuit proteome fraction on cellular metabolism, first, a list of up and downregulated genes was generated by comparing the ribosome density (RD) of each gene in the cell containing the circuit with that of the control cell. RD was used as the estimate of protein expression level. We calculated these differences for all eight circuit induction states. To do that, differential gene expression analysis was performed using EdgeR[134], where its 'exact test' analysis was performed in R (version 3.4.4) and adjusted p-values were calculated using built-in false discovery rate (FDR) correction. EdgeR requires biological replicates for its exact test analysis in order to estimate the dispersion in data (dispersion = BCV[2] where BCV is the biological coefficient of variation). A fixed BCV = 0.1 was used in all exact test analysis. Genes >2-fold change in expression and with p value and FDR less than 0.05 were selected as differentially expressed genes. These up and downregulated genes were used as the input for Gene Ontology analysis (GO)[135,136], where a GO annotation ID = 511145 was used for *E. coli*. GO identifies a list of regulated pathways based on their co-regulated gene members. Each regulated pathway was evaluated for potential relationship with estimated circuit proteome fraction across induction states. To do that, correlation between RD of each gene member in the pathway and the estimated circuit proteome fraction across induction states was calculated. Not all pathways were correlated. Only those pathways/genes with continuous decrease or increases in RD as a function of estimated circuit proteome fraction are reported in Supplementary Tables 2 and 3.

**Prediction of repressor off-target binding**. RD values of all native genes in *E. coli* were tested for correlation against the RD of individual repressors in the circuit (RD was used as the estimate of protein expression level). We were specifically searching for sigmoidal relationships, resembling off-target repression by repressors. To do that, for each repressor, the eight induction states of the circuit were divided into two subgroups, in which the repressor was either on or off. Next, for all the genes across the genome, RD values within each subgroup were averaged, resulting in an $RD_{ON,i}$ and $RD_{OFF,i}$ for each native gene $i$. We then calculated a fold-repression for each native gene $i$ using $RD_{OFF,i}$/$RD_{ON,i}$ and selected those native genes with >5-fold-repression for visual inspection of sigmoidal relationship. Only *hlyIIR* repressor showed significant off-target activity, with *aceB, tonB*, and *mipA* being the most repressed native genes. To identify potential binding site of *hlyIIR* repressor for its genomic off-targets, 100 nucleotides upstream or downstream (depending on the strand orientation) of the annotated TSS of each off-target gene was taken (upstream for *aceB* and *tonB* and downstream for *mipA*). The operator sequence of *hlyIIR* repressor (5'-ATATTTAAAATTCTTGTTTAAA-3') was then aligned to each DNA sequence using Needleman–Wunsch aligning algorithm[137] with mismatch penalty = 0, gap penalty = −2, and gap extension penalty = 0.

**Promoter motif identification**. Cryptic promoters were examined to determine whether they have motifs consistent with a σ70 promoter. First, the TSSs were identified (above), for which we found 228 forward and reverse cryptic promoters across the circuit DNA sequence and across the induction states. Those peaks

associated with ribozyme cleavage were manually removed. Promoter strength was calculated for each cryptic promoter using Eq. 3, and those promoters with activity >$10^{-5}$ RNAP/s per promoter were selected for motif identification (27 promoters passed the threshold, Supplementary Table 1), for which 50 nucleotides upstream of the TSS was taken for sequence analysis. Next, the −10 and −35 boxes of each promoter were manually identified. Not all promoters had both of the two boxes, and the spacing between them varied between 16 and 18 nucleotides. Finally, WebLogo (version 2.8.2)[138] was used to find common motifs across these promoter sequences (Supplementary Fig. 8).

**Measurement of cell growth**. Individual colonies of circuit containing *E. coli* cells were inoculated into 5 ml M9 minimal media with kanamycin and spectinomycin in a 14 ml culture tube (Corning, MA, 352059) and grown overnight for 16 h at 37 °C and 250 rpm in an Innova 44 shaker (Eppendorf, CT). The following day, cultures were diluted 5.6 μl into 1 ml M9 media with antibiotics and grown under the same condition for 3 h. The growth culture was then diluted 658-fold (45.5 μl into 30 ml) in 125 ml Erlenmeyer flasks (Pyrex) and different combinations of inducers were added. After 4 h of incubation where cells passed the lag phase and started the exponential phase of growth, 1 ml cell cultures were sampled intermittently and its $OD_{600}$ was read using a Cary 50 Bio spectrophotometer (Agilent, CA, 10068900) over the span of 6 hours. Cell growth rate μ was calculated as the slope of linear fit between log($OD_{600}$) (natural log) and time t before cultures entered stationary phase of growth. The doubling time $\tau = \log(2)/\mu$.

**Calculation of the gate's $K$ and $n$ parameters**. For each gate, the input and output promoter strengths (in units of RNAP/s) were obtained from RNA-seq data (Fig. 4a). From this data, the minimum and maximum output promoter strengths were assigned to $y_{min}$ and $y_{max}$, respectively, for use in Hill equation (Eq. 1). The Hill equation was then rearranged to log-linear format and optimal values for the dissociation constant *(K)* and cooperativity *(n)* parameters were found by Poisson regression using negative log-likelihood loss function. Calculations were performed in Python with the minimize function of scipy using the Nelder–Mead method since the loss function is convex. Each calculation was performed multiple times with varying initial guesses to verify convergence. For HlyIIR and BetI gates, the two outliers were excluded from fitting. $K$ was obtained in units of RNAP/s and $n$ is dimensionless. The fits for all 7 gates have been plotted in Supplementary Fig. 19. The fitted $K$ was then converted to the repressor binding constant $k$ in units of protein number using Eq. 22. Repressor binding constant $k$ and $n$ are presented in Table 2.

**Kinetic simulations of the genetic circuit**. A model is developed that tracks the mRNA and protein production over time and includes parameters associated with each genetic part, for which quantitative values can be extracted from the –omics data (Tables 1 and 2). The kinetics of protein dimerization and protein binding to DNA are assumed to be much faster than transcription, translation, and degradation, and thus are treated as being at pseudo-steady-state. In addition, the model does not take either RNAP or ribosome movement into account and assumes that every initiation event results in the production of a mRNA or protein, respectively. Thus, transcriptional attenuation and ribosomal pausing and early truncation are not taken into account. Second, there is no coupling between devices due to ribosomal and RNAP usage and no impact from the circuit on the host cell, including changing the growth rate. Note, however, that since the parameters are empirically extracted from –omics data involving the entire circuit, these effects will be accounted for to some extent in their values. Equations for the number of copies of the mRNA $m_i$ of gene $i$ are written below. Consider first the equation for the PhlF gate, which is located at the 5'-end of the circuit DNA sequence,

$$\frac{dm_{phlF}}{dt} = J_{phlF} - \gamma\big(\eta_{53}(1-b)+b\big)m_{phlF} \tag{6}$$

The RNAP flux from the two input promoters are summed as

$$J_{phlF} = y_{SrpR} + y_{BetI} \tag{7}$$

The degradation term includes the ribozyme cleavage efficiency $\eta_i \in [0,1]$ of the RiboJ(i) insulator. Uncleaved mRNA degrades more quickly, captured in the parameter $b = 2$, the value of which is obtained from the literature[139]. The mRNA degradation rate $\gamma = 0.0067$ s$^{-1}$ is also taken from the literature and assumed to be equal for all the circuit mRNAs[8,63]. The next gate on the circuit DNA has a similar form, except that it also has the possibility of RNAP read-through from the upstream gate (PhlF) due to an imperfect terminator. This can be written as

$$\frac{dm_{srpR}}{dt} = J_{srpR} - \gamma\big(\eta_{10}(1-b)+b\big)m_{srpR} \tag{8}$$

$$J_{srpR} = y_{BAD,1} + y_{Tet,1} + \frac{J_{phlF}}{T_{37}} \tag{9}$$

where the last term captures the read-through due to the terminator (Eq. 7). The terminator strength is $T_{37}$ (subscript is the last two digits of the terminator

name), the value of which is ratio of RNAPs that are blocked versus progressed[27]. The mRNAs produced by the remaining genes in the circuit are expressed as:

$$\frac{dm_{bm3R1}}{dt} = J_{bm3R1} - \gamma\big(\eta_{sarJ}(1-b)+b\big)m_{bm3R1} \tag{10}$$

$$J_{bm3R1} = y_{PhlF} + y_{HlyIIR} + \frac{J_{srpR}}{T_{00}} \tag{11}$$

$$\frac{dm_{betI}}{dt} = J_{betI} - \gamma\big(\eta_{57}(1-b)+b\big)m_{betI} \tag{12}$$

$$J_{betI} = y_{AmtR} + y_{AmeR} + \frac{J_{bm3R1}}{T_{2P11}} \tag{13}$$

$$\frac{dm_{ameR}}{dt} = J_{ameR} - \gamma\big(\eta_{54}(1-b)+b\big)m_{ameR} \tag{14}$$

$$J_{ameR} = y_{Tet,2} + \frac{J_{betI}}{T_{3P11}} \tag{15}$$

$$\frac{dm_{hlyIIR}}{dt} = J_{hlyIIR} - \gamma\big(\eta_{51}(1-b)+b\big)m_{hlyIIR} \tag{16}$$

$$J_{hlyIIR} = y_{Tac} + \frac{J_{ameR}}{T_{31}} \tag{17}$$

$$\frac{dm_{amtR}}{dt} = J_{amtR} - \gamma\big(\eta_{bydvJ}(1-b)+b\big)m_{amtR} \tag{18}$$

$$J_{amtR} = y_{BAD,2} + \frac{J_{hlyIIR}}{T_{36}} \tag{19}$$

where the flux terms $J_i$ capture all the transcription coming from upstream of the gate because they are all oriented in the same direction. Finally, the transcription of *yfp* from the circuit output promoter is written as

$$\frac{dm_{yfp}}{dt} = y_{BM3R1} - \gamma\big(\eta_{riboJ}(1-b)+b\big)m_{yfp} \tag{20}$$

The induction of the sensor and gate output promoters is assumed to be fast with respect to the repressor production or degradation[140]. The pseudo-steady-state approximation allows the response function of the gates to be used to calculate the activity of the output promoter as a function of the number of repressor proteins in the cell,

$$y_i = y_{\min,i} + \big(y_{\max,i} - y_{\min,i}\big)\frac{k_i^{n_i}}{k_i^{n_i} + R_i^{n_i}} \tag{21}$$

where $y_{\min,i}$, $y_{\max,i}$ and $n_i$ for gate $i$ are measured from the RNA-seq data (Tables 1 and 2, and Supplementary Fig. 19). The term $k_i$ is the binding constant of repressor $i$ to the output promoter of gate $i$ and is in units of protein number. This is related to the term $K_i$ used in the response function of gate $i$ (Supplementary Fig. 19, in units of RNAP/s), through

$$k_i = \left(\frac{\alpha_i}{\gamma\mu\big(\eta_i(1-b)+b\big)}\right)K_i \tag{22}$$

where $\alpha_i$ is the translation efficiency of the RBS controlling the translation of the repressor mRNA, $\mu = 0.00026\,\text{s}^{-1}$ is assumed to be dominated by cell division (the magnitude of which corresponds to a 45 min doubling time). Fits for the calculation of the $k_i$ for each gate are shown in Supplementary Fig. 19. Note that the parameters in Eq. 22 are not varied as part of the sensitivity analysis; rather, the simulation of the mutation of the output promoter is done by varying $k_i$. Finally, the number $R_i$ of protein $i$ in the cell can be calculated using simple mass action kinetics and the mRNA number:

$$\frac{dR_{phlF}}{dt} = \alpha_{P3}m_{phlF} - \mu R_{phlF} \tag{23}$$

$$\frac{dR_{srpR}}{dt} = \alpha_{S2}m_{srpR} - \mu R_{srpR} \tag{24}$$

$$\frac{dR_{bm3R1}}{dt} = \alpha_{B2}m_{bm3R1} - \mu R_{bm3R1} \tag{25}$$

$$\frac{dR_{betI}}{dt} = \alpha_{E1}m_{betI} - \mu R_{betI} \tag{26}$$

$$\frac{dR_{ameR}}{dt} = \alpha_{F1}m_{ameR} - \mu R_{ameR} \tag{27}$$

$$\frac{dR_{hlyIIR}}{dt} = \alpha_{H1}m_{hlyIIR} - \mu R_{hlyIIR} \tag{28}$$

$$\frac{dR_{amtR}}{dt} = \alpha_{A1}m_{amtR} - \mu R_{amtR} \tag{29}$$

$$\frac{dR_{yfp}}{dt} = \alpha_{64}m_{yfp} - \mu R_{yfp} \tag{30}$$

The equations are numerically solved using Python and the forward Euler method ($\delta t = 10\,\text{s}$) and typically solved to 12 hours. For each simulation trajectory, the equations are initiated by solving them until they reach steady-state under the initial states of the sensor inputs, noting that two sensors control two identical promoters in the circuit [$y_{Tac}$, ($y_{Tet,1}$, $y_{Tet,2}$), ($y_{BAD,1}$, $y_{BAD,2}$)]. At $t = 0$, a step change is applied to change the sensor promoters to their new activities. The on/off values used for the promoters are: 0.045/0.0000018 RNAP/s ($y_{Tac}$), 1.537/0.0000266 RNAP/s ($y_{Tet,1}$), 0.982/0.0000018 RNAP/s ($y_{Tet,2}$), 0.043/0.0000018 RNAP/s ($y_{BAD,1}$), 0.054/0.0000037 RNAP/s ($y_{BAD,2}$), also measured from RNA-seq data (Table 1). For the sensitivity analysis, one input variable was moved keeping all others at their nominal values. For each of 8 induction states, a full simulation trajectory (including reinitialization) was then run to 12 h with the new input variable. The YFP protein level at steady-state (end value of the simulation trajectory) was obtained for all eight induction states. The circuit by design has two on states (+/–/–, +/+/+) with IPTG/aTc/Ara induction, and six OFF states (the remaining combinations of inducers). The circuit score is calculated by taking the minimum $R_{YFP}$ of the two states that should be on, and dividing it by the maximum $R_{YFP}$ of the six states that should be off. This process was repeated with 1000 different values for each input variable, the variable was then returned to its nominal value before the sensitivity analysis was conducted for the next variable. Changes in circuit score as a function of each input variable are plotted in Fig. 5c; a circuit score less than 1 renders the circuit nonfunctional. A higher margin of circuit score = 3 was used as a cutoff for the plots. Parameters for the promoters are plotted for $y/N$, where $N = 9$ is the plasmid copy number for p15A to present in units of RNAP/s per DNA. For $P_{BM3R1}$, $N = 4$ is used due to the output promoter being on a different plasmid than the circuit with a higher copy number[18].

**Statistics and reproducibility.** For each statistical comparison, $R^2$ and $p$ value were calculated using corrcoef function in MATLAB (R2020a). When $p$ value <0.05, the corresponding $R^2$ is considered significant. Cell doubling times and fluorescence measurements were each repeated three times with similar results. For each induction state of the circuit, RNA-seq and ribosome profiling were performed once.

**Reporting summary.** Further information on research design is available in the Nature Research Reporting Summary linked to this article.

## Data availability

Source data are provided with this paper. FPKM, RD, and proteome fraction of all circuit and genomic proteins are listed in Supplementary Data 1. RNA-seq and ribosome profiling data collected in this study (both raw and processed) as well as the Supplementary Data 1 were deposited to Gene Expression Omnibus under the accession number GSE152664. EcoCyc database (https://ecocyc.org/)[133] was used to obtain a list of DNA-binding proteins in *E. coli* MG1655 genome, and DNA sequence of *E. coli* DH10B genome (NC_010473.1) was obtained from NCBI (https://www.ncbi.nlm.nih.gov/nuccore/NC_010473). Any other relevant data are available from the authors upon reasonable request.

## Code availability

Python scripts that implement the complete characterization of genetic circuits, sensitivity analysis, and dynamic model of genetic circuits are released as open-source software under the MIT license (GitHub repository: https://github.com/VoigtLab/Comprehensive_Genetic_Circuit_Analysis).

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

## Acknowledgements
This work was supported by US Defense Advanced Research Projects Agency (DARPA) 1KM grant HR0011-15-C-0084 (C.A.V., A.E.B., J.Z., and H.D.); US Department of Commerce—NIST (National Institute of Standards and Technology) award 70NANB16H164 (C.A.V., A.A.K.N., and J.Z.); DARPA SD2 grant (C.A.V., A.E.B., and H.D.); and US Department of Energy—Boulder award DE-SC0018368 (C.A.V. and H.D.). We would like to thank Professor Gene-Wei Li and Ariel Schieler at MIT Biology Department for their training on performing end-enriching RNA-seq and ribosome profiling experiments. We would also like to thank Alexander Cristofaro and Cassandra Bristol at the MIT-Broad Foundry for assisting in EdgeR and Gene Ontology analysis, and for assisting in the sequencing runs.

## Author contributions
J.Z., A.A.K.N., and C.A.V. conceived the study and designed the experiments. J.Z., A.E.B., and A.A.K.N. performed the experiments. A.E.B., J.Z., and H.D. performed the data analysis. A.E.B., J.Z., H.D., and C.A.V. wrote the manuscript.

## Competing interests
The authors declare no competing interests.
