## [Peer Review File · Nature Communications]

Reviewers' Comments:

Reviewer #1:

Remarks to the Author:

In this work, Borujeni and co-workers describe the use of RNA sequencing together with ribosome profiling to fully parameterize a regulatory circuit composed of 54 genetic parts. The authors found that, although the circuit overall performed as expected, several errors could be found at the molecular level. These errors reduce prediction accuracy and could lead to circuits to fail when the genetic parts are used in other designs.

This work builds upon a previous paper published by the same group (DOI: 10.15252/msb.20167461) but with significant improvements and new aspects that I consider relevant to the community and worth publishing in this journal. Namely, the authors used short RNA fragments and single-end mode sequencing to improve resolution in transcription analysis, and they use ribosome profiling to estimate for protein expression and to determine impacts in host metabolism. Moreover, these experiments were all combined in a single sequencing lane, decreasing experimental load and costs. Methods to extract the strengths of all genetic parts in the circuit are also developed.

Overall, the text is well written and structured, but I have some comments that should be addressed before acceptance.

Major comments:

1) For several times (lines 248, 286, 299 and 365), the authors state that the failures detected do not impact the circuit's function, referring to it using adverbs like "surprisingly", "interestingly", "serendipitously" and "remarkably". Nonetheless, they do not discuss why this might be; why does the overall response of the system is not affected despite complete gate failure? This is also related to how effective is the proposed diagnosis method for future circuit debugging in practice considering that so many parts not behaved as expected though the whole circuit may still/may not work as desired, i.e. it may not effectively/easily help identify the exact part(s) that contributed to a circuit failure.

2) Although the circuit functions as designed, given the results obtained, what would the authors change to make it perform "perfectly" at a molecular level? It would be interesting to see how the circuit behaves with an optimized design after part debugging - but given the current work limitations related to COVID-19, the authors should, at least, discuss this point.

3) The authors have studied the effect of the different induction rates of the circuit in host cells. They found several significant changes in the metabolism, but they don't go beyond enumerating those changes. What may be causing these changes; are they specific to this circuit or are they caused by circuit's proteins overexpression in general? What are the problems foreseen by the authors, if a larger circuit was introduced?

4) Overall, the "discussion" section should be significantly improved. Currently, it is too generalist and resembles an expanded abstract. The discoveries of this work should be further discussed instead of just being summarized in this section.

Minor comments:

Line 81: remove "repressor". Promoters are assigned to genes, not necessarily repressors. For instance, the output promoter controls reporter expression.

Line 86: "regulator" instead of "repressor". Although the authors built a repressor-based circuit, this paragraph is generalist and should reflect both repressors and activators.

Lines 104-107: please provide references for these sentences.

Supplementary Figure S5: please provide the full equation and R2 for the line fitting to the data.

Supplementary Figure S6: in panel b, because it is not a direct measure of protein levels (as stated in lines 105-107 and 354-355), terms "proteins" and "proteome" should be preceded by "estimate". Same is valid for other figures (e.g. Figure 5, Supplementary Figures S19 and S20) and text. Legend of Fig. S6, line 5 change "(a)" to bold.

Figure 2: Is the antisense cryptic promoter within P(BAD) promoters the native araC promoter?

Line 212: phIF also seem to have some degree of attenuation (Supplementary Figure S11).

Figure 2d and Supplementary Figure S11: change "No attenuation" text color to white to increase contrast.

Line 256: "riboJ5" should be "riboJ57"

Figures 5b and Supplementary Figure S19: the exact percentage for each fraction should be depicted in the figures to facilitate comparison. Is the native repressors fraction constant over the 8 conditions?

Figure 5c-f and Supplementary Figure S20: 9 points are plotted in each graph. What do they correspond to? According to Fig. 5c, they correspond to the different circuit states; if it is so, what is the 9th point?

Line 815: (f) in bold.

Line 821: Accession numbers are missing.

Line 374: Methods section

- Separate temperature units from numbers (section "Circuit induction").
- aTc concentration should be in mM similarly to the other inducers (Lines 394 and 411; Fig. 1 legend - Line 757).
- Under "Cello predictions", why didn't the authors use the new Eco1C2G2T2 UCF, previously published by the group (DOI: 10.15252/msb.20199401), to account for the non-additive promoter inputs and dynamics. Using this UCF will probably improve output predictions in Fig. 1c. Accordingly, consider revising sentence in lines 441-442 and panel b in Fig. 3.
- How were the RNA and cDNA fragments recovered from the acrylamide gels? Please provide, at least, a reference.

Line 841 and beyond: Too many references listed, while omitting some closely related ones in the field. For example, in the introduction/discussion when discussing previous work of applying RNA-seq for genetic circuit analysis, the below related ref should be included: "Orthogonality and burdens of heterologous AND gate gene circuits in E. coli", ACS Synthetic Biology 2018.

Reviewer #2:

Remarks to the Author:

Summary: In this manuscript, the authors investigate aberrant performance characteristics of a three-input logic circuit built on transcriptional NOR gates implemented in E. coli. Specifically, although the circuit executes more or less the correct logic with a fluorescent YFP output, a combination of RNA-seq and ribosome profiling show some unexpected complications with the circuit. These include antisense cryptic promoters and ribosome-binding sites, ineffective ribozymes, transcriptional attenuation, and even inversion of the expected circuit logic at a specific

internal gate. The authors carry out extensive characterization of part performance, including quantifying the part strength of all circuit components in parameters that actually possess physical meaning (e.g., RNAP flux through promoters). To perform their quantitative analyses, the authors implement a few clever strategies to remove RNA-seq biases present near promoters and terminators—this appears largely to give results that much intuitive understanding.

The authors do only study a single circuit in this work, but, given their purpose, I think this is fine. Overall, I found this work highly interesting and feel that it will be particularly relevant toward exploring the potential pitfalls of the design of highly complex synthetic gene circuits. I also want to praise the authors' highly effective data visualization—particularly in Figures 1 and 2—which makes it easy to understand a fairly complex data set. I recommend publication and have only a few minor comments that can likely be addressed by text changes:

- The final sentence of the manuscript's text argues that failed designs are due to failed "metrology" and not the "mystic of biology". Yet I am not sure that this work proves that. If anything, the authors show here that the metrology doesn't correlate with device performance because the parts themselves are flawed. The argument is that as a system grows more complex, part errors will lead to device malfunction, but I am not sure that better metrology is the only solution—it seems like part engineering (e.g., using only functioning ribozymes, refactoring genes to remove cryptic promoters/RBSs) is what we actually should be doing. I would like to see a more explicit and nuanced description of what ought to be the direct outcomes of this particular study, specifically outlining what work needs to be done to make biology easier to engineer.
- It is surprising to me how much the previous measurements of part strength (largely from fluorescence assays and performed in isolation) vary from the ones in this work, particularly the RBSs and terminators in Figure 3. In combination with the previous comment: I wonder if the authors can speculate on whether or not they expect these variations to manifest in the same way in other synthetic gene circuits with similar topologies, or if part strength really is this variable between contexts. How much would Cello's prediction of circuit function vary if, for instance, a single terminator's efficiency is misjudged by a factor of 10-100?
- The authors note in the methods section that tRNA copy numbers vary between induction conditions, though this data isn't experimentally shown anywhere (it is included in a discussion of mapping difficulties). It also isn't discussed in the main text when native genes that are up/down-regulated are enumerated. tRNA copy numbers could have corresponding effects on predicted ribosome occupancies—for instances, pause sequences could be read out not just due to "rare codons" (Line 221). I would like to see some discussion of this brought to the forefront (or perhaps some justification of why it isn't actually important).
- I found Figure S11 particularly interesting, especially because although transcriptional attenuation is generally low except for the transcripts pointed out, a few of the profiles are quite jagged. The biggest outlier here is amtR, which has a huge dip in mapped reads around nucleotide 150—despite the internal cryptic sense promoter. It appears these curves have been smoothed over, so that's unlikely to be a single transcript length that's problematic. Can the authors speculate on why these curves look the way they do? Is it a combination of TX attenuation and alternative TX initiation sites? Is it just noise/mapping inefficiencies? You can see this in Figure 1, too: there's an unexpected white line in the RNAP flux traces around this site in the traces where the amtR gate is ON.
- I understand the crux of Figure S10b intellectually, but the images are hard to understand. Can the authors re-factor that illustration to be clearer? Better labels of the schematic graphs would be helpful, for instance.

Typos:

Line 251, the word should be spelled RNase

Line 355, write out "sequencing" instead of seq

Table 1, there is an aberrant "1136" text in the middle of the first panel

Reviewer #1:

1. *For several times (lines 248, 286, 299 and 365), the authors state that the failures detected do not impact the circuit's function, referring to it using adverbs like "surprisingly", "interestingly", "serendipitously" and "remarkably". Nonetheless, they do not discuss why this might be; why does the overall response of the system is not affected despite complete gate failure? This is also related to how effective is the proposed diagnosis method for future circuit debugging in practice considering that so many parts not behaved as expected though the whole circuit may still/may not work as desired, i.e. it may not effectively/easily help identify the exact part(s) that contributed to a circuit failure.*

The robustness to gate failure is due to the topology of the network associated with this particular logic operation (Supplementary Figure S18). It can be understood simply from the Boolean functions of the gates. The NOR gate at the BetI position can fail in its logic operation without impacting the overall Boolean response of the circuit. This point is now clarified in the text.

2. *Although the circuit functions as designed, given the results obtained, what would the authors change to make it perform "perfectly" at a molecular level? It would be interesting to see how the circuit behaves with an optimized design after part debugging - but given the current work limitations related to COVID-19, the authors should, at least, discuss this point.*

We have added a paragraph to the discussion section that touches on this point.

3. *The authors have studied the effect of the different induction rates of the circuit in host cells. They found several significant changes in the metabolism, but they don't go beyond enumerating those changes. What may be causing these changes; are they specific to this circuit or are they caused by circuit's proteins overexpression in general? What are the problems foreseen by the authors, if a larger circuit was introduced?*

We have expanded our analysis and discussion of this point. Surprising to us, other than growth rate changes, we do not see differences between states that would be attributable to what has been described in the literature as "burden" (ribosome availability, induction of the heat shock response, etc). We do see metabolic shifts, as described in the text. Largely, we believe this is due to off-target effects, especially the SrpR repressor, where on the genome that impact central metabolism. Moving to larger circuits requires ensuring that regulators that are used are truly orthogonal to the host, although we do also expect generic "burden" to become more of an issue than we observe with this particular circuit.

4. *Overall, the "discussion" section should be significantly improved. Currently, it is too generalist and resembles an expanded abstract. The discoveries of this work should be further discussed instead of just being summarized in this section.*

The discussion section has been expanded and improved.

5. *Figures 5b and Supplementary Figure S19: the exact percentage for each fraction should be depicted in the figures to facilitate comparison. Is the native repressors fraction constant over the 8 conditions?*

The percentages for each fraction have been added to both Figures, as suggested. The native repressors fraction is relatively constant, varying between 2.4 and 2.9% across the eight conditions.

6. *Figure 5c-f and Supplementary Figure S20: 9 points are plotted in each graph. What do they correspond to? According to Fig. 5c, they correspond to the different circuit states; if it is so, what is the 9th point?*

The 9th data point is the control (cells containing the same plasmids backbone but lacking the circuit); the Figure caption has been edited to clarify this point.

7. *Supplementary Figure S6: in panel b, because it is not a direct measure of protein levels (as stated in lines 105-107 and 354-355), terms “proteins” and “proteome” should be preceded by “estimate”. Same is valid for other figures (e.g. Figure 5, Supplementary Figures S19 and S20) and text. Legend of Fig. S6, line 5 change “(a)” to bold.*

We added the term “estimated” in the text wherever we use RD, FPKM, or proteome fraction to represent gene expression.

8. *Under “Cello predictions”, why didn’t the authors use the new Eco1C2G2T2 UCF, previously published by the group (DOI: 10.15252/msb.20199401), to account for the non-additive promoter inputs and dynamics. Using this UCF will probably improve output predictions in Fig. 1c. Accordingly, consider revising sentence in lines 441-442 and panel b in Fig. 3.*

The construction of this circuit and the acquisition of the sequencing data predates the new version of Cello and associated UCFs.

9. *Minor points:*

- *Line 81: remove “repressor”. Promoters are assigned to genes, not necessarily repressors. For instance, the output promoter controls reporter expression.*
- *Lane 86: “regulator” instead of “repressor”. Although the authors built a repressor-based circuit, this paragraph is generalist and should reflect both repressors and activators.*
- *Lines 104-107: please provide references for these sentences.*
- *Supplementary Figure S5: please provide the full equation and R2 for the line fitting to the data.*
- *Figure 2d and Supplementary Figure S11: change “No attenuation” text color to white to increase contrast.*
- *Line 256: “riboJ5” should be “riboJ57”*

- Line 815: *(f)* in bold.

All of the edits above were implemented as suggested.

- Figure 2: *Is the antisense cryptic promoter within P(BAD) promoters the native araC promotor?*

This antisense cryptic promoter is located close to the native *araC* promoter P_C , specifically, at the 5'-end of the O_{1L} site of the P_{BAD} promoter. We have clarified this in the text.

- Line 212: *phIF also seem to have some degree of attenuation (Supplementary Figure S11).*

While there is some variability in the *phIF* transcriptional profile, we do not see the continuous decline associated with attenuation. We have edited the text to better define the metric used to determine whether attenuation is a significant contributor to the profile.

- Line 821: *Accession numbers are missing.*

The GEO accession number is added to the text. It includes all processed RNA-seq and ribosome profiling profiles, as well as tables of FPKM, RD, and proteome fractions. There has been a delay in obtaining the raw sequencing FASTQ files but they will be added before publication.

- Line 841 and beyond: *Too many references listed, while omitting some closely related ones in the field. For example, in the introduction/discussion when discussing previous work of applying RNA-seq for genetic circuit analysis, the below related ref should be included: "Orthogonality and burdens of heterologous AND gate gene circuits in E. coli", ACS Synthetic Biology 2018.*

We have updated our reference list, including the above suggested reference.

- Line 374: *Methods section - Separate temperature units from numbers (section "Circuit induction"). - aTc concentration should be in mM similarly to the other inducers (Lines 394 and 411; Fig. 1 legend - Line 757). - How were the RNA and cDNA fragments recovered from the acrylamide gels? Please provide, at least, a reference.*

All of the suggested changes were made, and the Methods has been expanded to describe the RNA and cDNA gel recovery steps.

Reviewer #2:

1. *The final sentence of the manuscript's text argues that failed designs are due to failed "metrology" and not the "mystic of biology". Yet I am not sure that this work proves that. If anything, the authors show here that the metrology doesn't correlate with device performance because the parts themselves are flawed. The argument is that as a system grows more complex, part errors will lead to device malfunction, but I am not sure that better metrology is the only solution—it seems like part engineering (e.g., using only functioning ribozymes, refactoring genes to remove cryptic promoters/RBSs) is what we actually should be doing. I would like to see a more explicit and nuanced description of what ought to be the direct outcomes of this particular study, specifically outlining what work needs to be done to make biology easier to engineer.*

We agree and have edited the text to expand beyond metrology as necessary for design.

2. *It is surprising to me how much the previous measurements of part strength (largely from fluorescence assays and performed in isolation) vary from the ones in this work, particularly the RBSs and terminators in Figure 3. In combination with the previous comment: I wonder if the authors can speculate on whether or not they expect these variations to manifest in the same way in other synthetic gene circuits with similar topologies, or if part strength really is this variable between contexts. How much would Cello's prediction of circuit function vary if, for instance, a single terminator's efficiency is misjudged by a factor of 10-100?*

We agree with this comment and decided a more rigorous analysis would be interesting. This has been added as a new Figure 5. In short, we extracted all the part parameters from the ribosome-profiling and RNA-seq datasets (RBS efficiencies, terminator strengths, etc) and then used these parameters to inform an ODE model of part function. Not surprisingly, this is able to recapitulate the circuit function and predict the dynamics. Then, we define a metric for whether the circuit is performing the correct logic and varied each of the parameters individually... inspired by Gary O'Dell's segmentation work. This has really interesting behavior and shows that many parts can have extremely wide ranges of activity and still lead to a functional circuit. It provides tolerances on the ranges of part parameters that are acceptable, with some being fragile and others robust with respect to this particular circuit topology.

3. *The authors note in the methods section that tRNA copy numbers vary between induction conditions, though this data isn't experimentally shown anywhere (it is included in a discussion of mapping difficulties). It also isn't discussed in the main text when native genes that are up/down-regulated are enumerated. tRNA copy numbers could have corresponding effects on predicted ribosome occupancies—for instances, pause sequences could be read out not just due to "rare codons" (Line 221). I would like to see some discussion of this brought to the forefront (or perhaps some justification of why it isn't actually important).*

Clarifying text has been added. Indeed, there is a lot of variability between samples (15% to 65% of the RNA is tRNA). Unfortunately, we have not been able to interpret these numbers with respect to the impact on the circuit. Although the end-enriching method can capture most of tRNA species with their 5'- and 3'-ends, we found that it is not an accurate method for all tRNAs species, as some of tRNAs appear to be partially degraded, or missing during ribosomal RNA removal steps. We also did not find correlation between codon usage in ribosome profiling and tRNA copy number for codons with high ribosome pausing. We should note that our repressor genes have been codon optimized to remove rare codons, so tRNA copy number is not expected to be significantly impacted.

4. *I found Figure S11 particularly interesting, especially because although transcriptional attenuation is generally low except for the transcripts pointed out, a few of the profiles are quite jagged. The biggest outlier here is amtR, which has a huge dip in mapped reads around nucleotide 150—despite the internal cryptic sense promoter. It appears these curves have been smoothed over, so that's unlikely to be a single transcript length that's problematic. Can the authors speculate on why these curves look the way they do? Is it a combination of TX attenuation and alternative TX initiation sites? Is it just noise/mapping inefficiencies? You can see this in Figure 1, too: there's an unexpected white line in the RNAP flux traces around this site in the traces where the amtR gate is ON.*

The dip at the beginning of *amtR* was unexpected and has puzzled us as well. These fluctuations are consistent across all 8 induction states of the circuit, especially when the gene is expressed, which points away from it being an effect due to noise. Mapping bias may be the cause of this effect. There is a strong hairpin immediately upstream of the dip position, but we do not know how this could lead to this effect.

5. *I understand the crux of Figure S10b intellectually, but the images are hard to understand. Can the authors re-factor that illustration to be clearer? Better labels of the schematic graphs would be helpful, for instance.*

The Supplementary Figure S10 has been revised.

6. *Typos:*

Line 251, the word should be spelled RNase

Line 355, write out "sequencing" instead of seq

Table 1, there is an aberrant "1136" text in the middle of the first panel

These typos have been corrected.

Reviewers' Comments:

Reviewer #1:

Remarks to the Author:

The authors have addressed all the issues raised and this reviewer has no further comments, though it would be better to have highlighted the changes in the revised manuscript.

Reviewer #2:

Remarks to the Author:

The authors have addressed my concerns and this is an impactful manuscript.